# Proteome-wide Mendelian randomization in global biobank meta-analysis reveals multi-ancestry drug targets for common diseases

## Graphical abstract

## Authors

Huiling Zhao, Humaria Rasheed, Therese Haugdahl Nøst, ..., Benjamin M. Neale, Tom R. Gaunt, Jie Zheng

## Correspondence

wzhou@broadinstitute.org (W.Z.), bneale@broadinstitute.org (B.M.N.), tom.gaunt@bristol.ac.uk (T.R.G.), jie.zheng@bristol.ac.uk (J.Z.)

## In brief

We present a multi-ancestry proteome-wide MR pipeline from the Global Biobank Meta-analysis Initiative (GBMI) to identify protein-disease pairs with putative causal evidence in multi-ancestries and with ancestry-specific effects. By integrating the MR signals with observational and clinical trial evidence, we evaluate the efficacy of existing drugs and identify drug-repurposing opportunities.

## Highlights

- A multi-ancestry proteome-wide Mendelian randomization (MR) analysis pipeline

- Identify putative protein-disease pairs in European and African ancestries

- Identify ancestry-specific causal effects

- Prioritize new drug targets and drug-repurposing opportunities

 Zhao et al., 2022, Cell Genomics 2, 100195
November 9, 2022 © 2022 The Author(s).

CellPress

# Proteome-wide Mendelian randomization in global biobank meta-analysis reveals multi-ancestry drug targets for common diseases

Huiling Zhao,[1] Humaria Rasheed,[1,2,3] Therese Haugdahl Nøst,[2,4] Yoonsu Cho,[1] Yi Liu,[1] Laxmi Bhatta,[2] Arjun Bhattacharya,[5,6] Global Biobank Meta-analysis Initiative, Gibran Hemani,[1] George Davey Smith,[1,7] Ben Michael Brumpton,[3,8,9,15] Wei Zhou,[10,11,12,15,*] Benjamin M. Neale,[11,12,15,*] Tom R. Gaunt,[1,7,15,*] and Jie Zheng[1,13,14,15,16,*]

[1]MRC Integrative Epidemiology Unit (IEU), Bristol Medical School, University of Bristol, Oakfield House, Oakfield Grove, Bristol BS8 2BN, UK
[2]K.G. Jebsen Center for Genetic Epidemiology, Department of Public Health and Nursing, NTNU, Norwegian University of Science and Technology, Trondheim, Norway
[3]Division of Medicine and Laboratory Sciences, University of Oslo, Oslo, Norway
[4]Department of Community Medicine, UIT The Arctic University of Norway, 9037 Tromsø, Norway
[5]Department of Pathology and Laboratory Medicine, David Geffen School of Medicine, University of California, Los Angeles, CA, USA
[6]Institute of Quantitative and Computational Biosciences, David Geffen School of Medicine, University of California, Los Angeles, CA, USA
[7]NIHR Bristol Biomedical Research Centre, Bristol, UK
[8]HUNT Research Center, Department of Public Health and Nursing, NTNU, Norwegian University of Science and Technology, 7600 Levanger, Norway
[9]Clinic of Medicine, St. Olavs Hospital, Trondheim University Hospital, Trondheim, Norway
[10]Department of Internal Medicine, Division of Cardiovascular Medicine, University of Michigan, Ann Arbor, MI 48109, USA
[11]Analytic and Translational Genetics Unit, Massachusetts General Hospital, Boston, MA 02114, USA
[12]Stanley Center for Psychiatric Research, Broad Institute of Harvard and MIT, Cambridge, MA 02142, USA
[13]Department of Endocrine and Metabolic Diseases, Shanghai Institute of Endocrine and Metabolic Diseases, Ruijin Hospital, Shanghai Jiao Tong University School of Medicine, Shanghai, China
[14]Shanghai National Clinical Research Center for Metabolic Diseases, Key Laboratory for Endocrine and Metabolic Diseases of the National Health Commission of the PR China, Shanghai Key Laboratory for Endocrine Tumor, State Key Laboratory of Medical Genomics, Ruijin Hospital, Shanghai Jiao Tong University School of Medicine, Shanghai, China
[15]These authors contributed equally
[16]Lead contact
*Correspondence: wzhou@broadinstitute.org (W.Z.), bneale@broadinstitute.org (B.M.N.), tom.gaunt@bristol.ac.uk (T.R.G.), jie.zheng@bristol.ac.uk (J.Z.)

## SUMMARY

Proteome-wide Mendelian randomization (MR) shows value in prioritizing drug targets in Europeans but with limited evidence in other ancestries. Here, we present a multi-ancestry proteome-wide MR analysis based on cross-population data from the Global Biobank Meta-analysis Initiative (GBMI). We estimated the putative causal effects of 1,545 proteins on eight diseases in African (32,658) and European (1,219,993) ancestries and identified 45 and 7 protein-disease pairs with MR and genetic colocalization evidence in the two ancestries, respectively. A multi-ancestry MR comparison identified two protein-disease pairs with MR evidence in both ancestries and seven pairs with specific effects in the two ancestries separately. Integrating these MR signals with clinical trial evidence, we prioritized 16 pairs for investigation in future drug trials. Our results highlight the value of proteome-wide MR in informing the generalizability of drug targets for disease prevention across ancestries and illustrate the value of meta-analysis of biobanks in drug development.

## INTRODUCTION

The efficacy of drugs is typically evaluated in one or a small number of populations during phase 3 clinical trials. Utilizing multi-ancestry data for drug discovery has several obvious benefits. First, ancestry-specific data may lead to novel drug target discovery. For example, the loss-of-function mutation in PCSK9 was first identified using sequencing data from African ances-

tries.[2] Second, genetic tools provide a cost-effective way to understand whether the drug targets may have similar or different effects across ancestries, which could help to improve the generalizability of drug interventions across ancestries.

One possible approach is to use genetic variants that influence the drug target as proxies to cost-effectively predict treatment response.[3,4] Proteome-wide Mendelian randomization (MR) utilizes genetic predictors of protein levels to test the putative

causal effects of proteins on common diseases. Subject to the key assumptions, MR can provide evidence of the putative causal roles of thousands of proteins on risk of a wide range of diseases.[5–7] Some recent proteome studies have built proteome-phenome maps in samples of up to 35,559 European participants[8,9] and further suggested that drug targets with robust MR and colocalization evidence are more likely to be successful in drug trials.[6] Moving beyond these studies, MR could play a key role in prioritizing drug targets in different ancestries,[10] which may inform the design of future trials.[11]

Multi-ancestry studies are gaining increasing prominence because of the importance of understanding differences in disease etiology between ancestries. Others have developed and applied trans-ancestry methods for genetic correlation,[12] polygenic risk score,[13,14] and fine mapping[15,16] analyses. However, multi-ancestry putative causal inference using MR is still in its infancy.[17] One major issue restricting the development of multi-ancestry MR is the unbalanced representation of genome-wide association study (GWAS) samples across different ethnic groups, with one commentary reporting that across published GWASs, 78% of participants were of European ancestry.[18] Consequently, most published proteome GWAS and MR studies have been restricted to European ancestry.[5–8,19,20] This bias in population coverage causes two issues: (1) we lack sufficient proteomic GWASs in non-European ancestries, which restricts our ability to identify protein quantitative trait loci (pQTLs) in other ancestries; and (2) without well-powered disease GWASs in non-European ancestries, we have little opportunity to identify multi-ancestry and ancestry-specific protein-disease associations.

A recent plasma proteome GWAS has identified pQTLs in both European and African ancestries and compared the genetic architecture of the proteome across ancestries.[21] This study further estimated the associations of proteins on plasma urate and gout in Europeans using a previously described transcriptome-wide association study pipeline.[22] The integration of this unique multi-ancestry pQTL resource with ancestry-enriched GWAS resources within the Global Biobank Meta-analysis Initiative (GBMI) has presented an ideal opportunity for a multi-ancestry proteome MR analysis. GBMI has collated a multi-ancestry genetic dataset with 2.6 million subjects, including samples from Asian, African, Hispanic American, and European ancestries; has standardized phenotype/disease definitions; and has applied a universal GWAS analysis pipeline across these biobanks.[23] This initiative has enabled us to conduct a multi-ancestry proteome MR using well-harmonized GWAS data.

In this study, we systematically estimated the putative causal role of 1,311 and 1,310 proteins, measured in populations from African and European ancestry, respectively (1,076 proteins in both ancestries), on eight complex diseases using a comprehensive ancestry-specific MR pipeline based on our previous approach in European datasets.[6] We further estimated the consistency of pQTLs across ancestries, identified potential multi-ancestry and ancestry-specific putative causal proteins that prevent disease onset, and integrated MR findings with observational and clinical trial evidence[24] to prioritize drug targets. We report our results in an openly accessible database: EpiGraphDB[1] (https://epigraphdb.org/multi-ancestry-pwmr/).

## RESULTS

### Summary of selection and validation of protein and disease data

*cis*-acting pQTLs within 500 kb of the protein-coding gene were selected as genetic instruments for the proteome MR analyses because *cis*-acting pQTLs are more likely to have protein-specific effects than *trans*-acting pQTLs.[5] For the discovery MR analysis, 6,144 conditionally independent pQTLs of 1,310 proteins in 7,213 Europeans (Table S1) and 3,875 conditionally independent pQTLs of 1,311 proteins in 1,871 Africans (Table S2) derived from the Atherosclerosis Risk in Communities Study (ARIC) study[21] were selected as candidate genetic instruments for their respective proteins. We defined conditionally independent pQTLs as a set of *cis*-acting pQTLs that have independent genetic effects on the tested protein. The conditionally independent pQTLs were identified using the forward-backward stepwise regression implemented in the QTLtools package (https://qtltools.github.io/qtltools/), followed by the fine mapping method SuSiE.[25] To increase reliability and boost power, we developed a three-step instrument validation pipeline to filter the pQTLs that best fit the MR assumptions. First, to avoid the potential issue of collinearity of the MR model, we applied linkage disequilibrium (LD) clumping to remove pQTLs strongly correlated with each other ($r^2 < 0.6$). Second, we estimated the instrument strength using F-statistics, excluding pQTLs with F-statistics lower than 10 from the MR analysis to avoid potential weak instrument bias.[26] Third, we applied the MR Steiger filtering approach[27] to exclude pQTLs with potential reverse causality[28] (i.e., where genetic predisposition to disease has a putative causal effect on the protein). After selection, a total of 3,550 pQTLs of 1,311 proteins in Africans and 5,418 pQTLs of 1,310 proteins in Europeans were selected as instruments for the MR analysis (Figure 1). We further divided the pQTLs into three tiers using a refined instrument validation process we previously developed[6] (details are in Figures S1 and S2, Table S3, and STAR Methods). We kept all instruments for the MR analysis but annotated our results with these tiers and recommend that results from heterogeneous and non-specific instruments be treated with caution.

For the validation MR analysis, 289 conditionally independent pQTLs of 289 proteins in up to 6,000 Europeans were selected from Zheng et al.[6] (Table S4), and 290 conditionally independent pQTLs of 290 proteins in 467 Africans from the African American Study of Kidney Disease and Hypertension Cohort Study (AASK) cohort (Table S5) were selected as instruments for the validation MR analysis (Figure 1).

For the outcomes of the MR analysis, we selected 8 of the 14 diseases from GBMI on the basis that we had full GWAS summary statistics in both European and African ancestries and relatively good statistical power (more than 100 cases). The eight disease outcomes included idiopathic pulmonary fibrosis (IPF), primary open-angle glaucoma (POAG), heart failure (HF), venous thromboembolism (VTE), stroke, gout, chronic obstructive pulmonary disease (COPD), and asthma (Tables S6A and S6B).

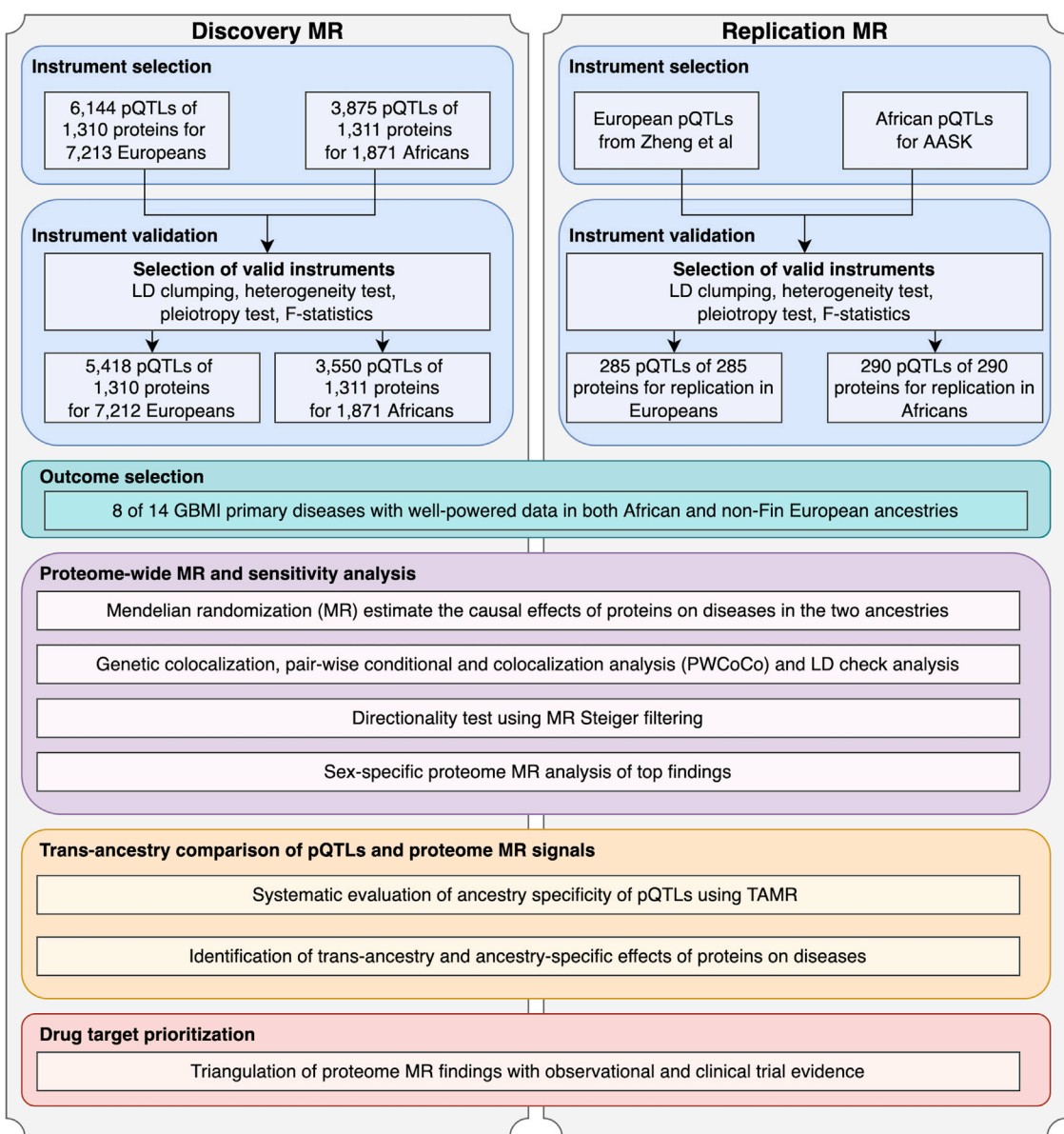

**Figure 1. Study design of the multi-ancestry proteome-wide Mendelian randomization (MR) in Global Biobank Meta-analysis Initiative**

## Estimation of putative causal effects of proteins on diseases in African and European ancestries

We undertook two-stage (discovery and validation) MR and sensitivity analyses to systematically evaluate evidence for the putative causal effects of 1,311 plasma proteins on the eight diseases in African ancestry and separately for 1,310 proteins on the same eight diseases in European ancestry. Of note, because the outcome data we used here are from case-control GWASs, the protein-disease pairs being identified in this study are potential targets for disease prevention rather than treatment. Of these proteins, 1,076 of them had instruments in both ancestries (Tables S1 and S2). 544 of them (20.8%) have only one pQTL, 599 (22.9%) have two pQTLs, and 1,478 (56.4%) have three or more pQTLs in the *cis* region. For proteins with

one pQTL, we applied the Wald ratio test.[29] For proteins with two or more pQTLs, we applied a generalized inverse variance weighted approach (IVW; gIVW),[30] which takes into account the LD correlation between nearby *cis* instruments and increases the reliability of the MR analysis (because conditional independent pQTLs could still be in LD, a conventional IVW may double-count effects among these signals). For proteins with three or more pQTLs, we further applied a generalized MR-Egger regression (gEgger) approach,[30,31] which allowed us to estimate the potential effect of pleiotropy on the MR estimates. When a certain pQTL was missing in the disease GWAS data, we used a proxy genetic variant in high LD with that pQTL ($r^2 > 0.8$ in the 1000 Genomes data for the relevant population[32]) instead (Figure 1).

## Discovery MR and sensitivity analyses

We conducted discovery MR on 10,318 protein-disease pairs in European ancestry and 9,858 pairs in African ancestry. Among these, 69 MR signals in European ancestry and 1 signal in African ancestry reached a Benjamini-Hochberg false discovery rate (FDR) of 0.05. To maximize the possibility of identifying true putative causal effects, we conducted a range of sensitivity analyses on the MR signals that passed the FDR threshold, which included a pleiotropy test using the gEgger intercept term,[31] a heterogeneity test using Cochran's Q for gIVW analysis and Rücker's Q for gEgger analysis,[33,34] and a set of genetic colocalization analyses (including conventional colocalization, pairwise conditional and colocalization [PWCoCo], and LD check).[6,35] The test of gEgger intercept suggested that 10 of the 69 (14.5%) potential protein-disease effects in Europeans and none of the effects in Africans showed evidence of being influenced by directional pleiotropy (Table S9A). Because pleiotropy invalidates the exclusion restriction assumption of MR, these 10 results were excluded from the candidate putative causal effects list. A total of 59 MR signals in European ancestry and 1 in African ancestry were therefore considered as candidate protein-disease pairs in the discovery analysis (Tables S7A and S8A).

For the remaining 59 protein-disease pairs in European ancestry and 1 pair in African ancestry, we were able to conduct a heterogeneity test on 53 of them. 38 (71.7%) showed little evidence of heterogeneity (Tables S7A and S8A). This observation implies that the conditionally independent *cis* pQTLs of the same protein tend to show proportionally similar effects on the relevant disease outcomes. Because heterogeneity could be caused by various factors,[11] we kept the MR signals with evidence of heterogeneity in the candidate list but annotated this in our results.

To further distinguish putative causal protein-disease pairs from confounding by LD, we applied three colocalization approaches: conventional colocalization,[35] PWCoCo, and LD check analysis.[6] The LD check analysis, which estimated the LD between each pQTL and disease-associated GWAS signal in the *cis* region, suggested that 43 protein-disease pairs in European ancestry and one pair in African ancestry showed evidence of approximate colocalization (pairwise LD $r^2 > 0.7$; Tables S7A and S8A). This includes protein levels of ABO, which showed robust MR and LD check evidence on VTE, with the same direction of effect in both European and African ancestries (odds ratio [OR] in Europeans = 1.11, p = $5.59 \times 10^{-11}$, LD $r^2 = 1$; OR in Africans = 1.33, p = $2.82 \times 10^{-6}$; LD $r^2 = 0.80$; Tables S7 and S8). The conventional colocalization and PWCoCo showed colocalization evidence for 18 and 1 protein-disease pairs in European and African ancestries, respectively (colocalization posterior probability > 70%). For example, we identified the effect of protein level of PROC on VTE using a *trans*-acting variant in the PROCR region in our previous proteome-wide MR study,[6] which was confirmed using the same variant using the GBMI VTE GWAS data.[36] In this study, we estimated the same effect of PROC level on VTE in European ancestry using *cis*-acting pQTLs (p = $1.45 \times 10^{-8}$, colocalization probability = 99%) but weaker evidence in African ancestry (p = $8.96 \times 10^{-3}$, colocalization probability = 11%). In summary, 46 of 60 (76.7%) MR signals in European and/or African ancestries showed colocalization and/or LD check evidence (Tables S7A and S8A).

Finally, we considered potential aptamer-binding artifacts driven by protein-altering variants within the target sequence. Among the 5,418 and 3,550 pQTLs selected in European and African ancestries, 1,421 (15.8%) of the pQTLs or their LD proxies ($r^2 > 0.8$ in 1000 Genome reference panel) were annotated as missense, stop-lost, or stop-gained variants using Ensembl Variant Effect Predictor (VEP)[37] (Tables S1 and S2). For MR signals using these pQTLs as genetic instruments, we flagged the MR effect estimates and recommend caution in their interpretation. Among the 59 robust European MR signals and 1 robust African MR signal, 26 (43.3%) were estimated using non-coding variants (and for which LD proxies were also non-coding variants) as the instrument in European and/or African ancestries (Tables S7 and S8) and are therefore not likely to be influenced by aptamer-binding artifacts.

As summarized in Table 1, 45 MR signals in European ancestry and 1 signal in African ancestry passed the FDR-corrected threshold and showed colocalization/LD check evidence and little evidence of pleiotropy.

## Validation of MR signals using pQTL data from independent samples

We selected the 59 European MR signals and 1 African signal that passed the FDR threshold of 0.05 for the validation MR analysis (Tables S7A and S8A). The conditional independent pQTLs were selected as instruments from two non-overlapped studies, Zheng et al.[6] and AASK.[21] After instrument selection and validation, we were able to conduct validation MR in 28 pairs and 1 pair in European and African ancestries, respectively. Among these pairs, 14 protein-disease pairs and 1 protein-disease pair showed MR evidence (FDR < 0.05 in validation analysis) in European and African ancestries, respectively (Tables S10A and S10B). When data were available, we applied the same sensitivity analyses, including colocalization analysis, for the validation MR signals. Six pairs showed colocalization evidence in European ancestry (Tables S10A and S10B).

## Sex-specific MR analysis

The treatment response of drugs often differs by sex.[38] To investigate the potential influence of sex-specific genetic effects on our proteome MR signals, we conducted sex-specific proteome MR using male- and female-only disease GWASs provided by GBMI (Table S6). Among the protein-disease pairs, 7,498 protein-disease pairs in European ancestry and 7,693 pairs in African ancestry have available data to conduct the MR analysis in both females and males. The pairwise Z score test comparing the male- and female-only MR estimate was applied to identify protein-disease pairs with sex-specific effects. After applying an FDR threshold of 0.05 for the pairwise Z score p values in European and African ancestries, 12 protein-disease pairs in European ancestry and 3 pairs in African ancestry showed robust evidence of difference in MR estimates between sexes (Table S11). Among these protein-disease pairs, three of them were related to proteins of existing drug targets, which included interleukin-17 receptor antagonist (IL-17RA) level on asthma, endoplasmic reticulum aminopeptidase 1 (ERAP1) level (target of tosedostat) on IPF, and NQO1 level (target of vatiquinone) on HF (Figure 2). The protein IL-17RA is a target of the drug brodalumab, and

**Table 1. Summary of proteome-wide MR results in European and African ancestries, related to STAR Methods**

| MR estimate of effect on disease | Cases/controls | | MR signals with FDR < 0.05 and colocalization evidence | |
| --- | --- | --- | --- | --- |
| | European ancestry | African ancestry | European ancestry | African ancestry |
| Idiopathic pulmonary fibrosis (IPF) | 4,066/751,962 | 168/8,364 | 3 | 0 |
| Primary open-angle glaucoma (POAG) | 11,657/957,371 | 483/26,323 | 3 | 0 |
| Heart failure (HF) | 27,064/796,602 | 970/16,823 | 0 | 0 |
| Venous thromboembolism (VTE) | 15,907/696,395 | 1,031/16,743 | 17 | 1 |
| Stroke | 15,520/852,432 | 1,161/24,411 | 2 | 0 |
| Gout | 20,512/843,507 | 1,313/33,896 | 4 | 0 |
| Chronic obstructive pulmonary disease (COPD) | 51,231/800,577 | 1,978/27,699 | 2 | 0 |
| Asthma | 100,736/1,219,215 | 5,054/2,7599 | 14 | 0 |
| Total MR results | | | 45 | 1 |
| Multi-ancestry and ancestry-specific MR results | Cases/controls | | Ancestry-specific MR signals with FDR < 0.05 | |
| | European ancestry | African ancestry | European ancestry | African ancestry |
| Idiopathic pulmonary fibrosis (IPF) | 4,066/751,962 | 168/8,364 | 4 | 0 |
| Primary open-angle glaucoma (POAG) | 11,657/957,371 | 483/26,323 | 9 | 1 |
| Heart failure (HF) | 27,064/796,602 | 970/16,823 | 7 | 1 |
| Venous thromboembolism (VTE) | 15,907/696,395 | 1,031/16,743 | 18 | 1 |
| Stroke | 15,520/852,432 | 1,161/24,411 | 4 | 3 |
| Gout | 20,512/843,507 | 1,313/33,896 | 9 | 2 |
| Chronic obstructive pulmonary disease (COPD) | 51,231/800,577 | 1,978/27,699 | 10 | 2 |
| Asthma | 100,736/1,219,215 | 5,054/2,7599 | 28 | 2 |
| Total MR results | | | 89 | 12 |

FDR, false discovery rate; MR, Mendelian randomization.

the efficacy of this drug on asthma was tested in a phase 2 trial of 421 participants (brodalumab versus placebo; ClinicalTrials.gov: NCT01902290). However, the trial was terminated because of lack of efficacy. Our MR results showed that genetically increased IL-17RA level was associated with increased asthma risk in males (OR = 1.03; 95% confidence interval [CI] = 1.02–1.04; p = 1.69 × 10⁻⁹) but showed little effect in females (OR = 1.00; 95% CI = 1.00–1.01; p = 0.35) in European ancestry. Although the protective effect of IL-17RA inhibition on asthma was relatively minor in the sex-combined trial (ClinicalTrials.gov: NCT01902290), our study suggested that the efficacy of this target on asthma in males may be worth reconsideration in future trials.

### Multi-ancestry comparison of pQTLs and proteome MR signals
#### Systematic evaluation of ancestry specificity of pQTLs
pQTLs may show different effects across ancestries because of differing allele frequencies, LD structure, sample sizes, or interactions. We systematically evaluated the ancestry specificity of pQTLs using the TAMR package (https://github.com/universe77/TAMR). TAMR allows us to estimate how often pQTLs across two ancestries have a substantially different direction of effect or different level of statistical evidence compared with expectation, where the expected degree of replication was calculated using a Bayesian winner's curse correction described in a previous study.[39] Using the expected degree of

replication as a benchmark offers a comparison of pQTLs that takes into account the differences in allele frequency, sample size, and effect size of pQTLs across ancestries. The 1,076 proteins with full summary statistics and pQTL signals in both ancestries were included in this analysis.

We first estimated how often African pQTLs showed ancestry-specificity/consistency compared with European pQTLs. Among the expected pQTLs, 73.0% of them show consistent direction of effect across the two ancestries. 83.6% of the expected pQTLs were observed to reach the GWAS genome-wide evidence threshold (p < 5 × 10⁻⁸) in both ancestries (Table S12A). Conversely, we estimated how often European pQTLs showed ancestry specificity/consistency compared with African pQTLs. In agreement with the results with African pQTLs, we found that 71.8% of the expected pQTLs showed a consistent direction of effect across the two ancestries. However, only 60.8% of the expected pQTLs met the GWAS genome-wide evidence threshold in both ancestries largely driven by statistical power differences (Table S12B).

### Identification of multi-ancestry and ancestry-specific pQTLs
Ancestry specificity of pQTL instruments can be a direct cause of differences in estimated ancestry-specific putative causal effects of proteins on diseases. Therefore, there is a need to distinguish trans-ancestry and ancestry-specific pQTLs among our pQTL instruments. We stratified the 1,310 and 1,311 tested

**Cell Genomics**
**Article**

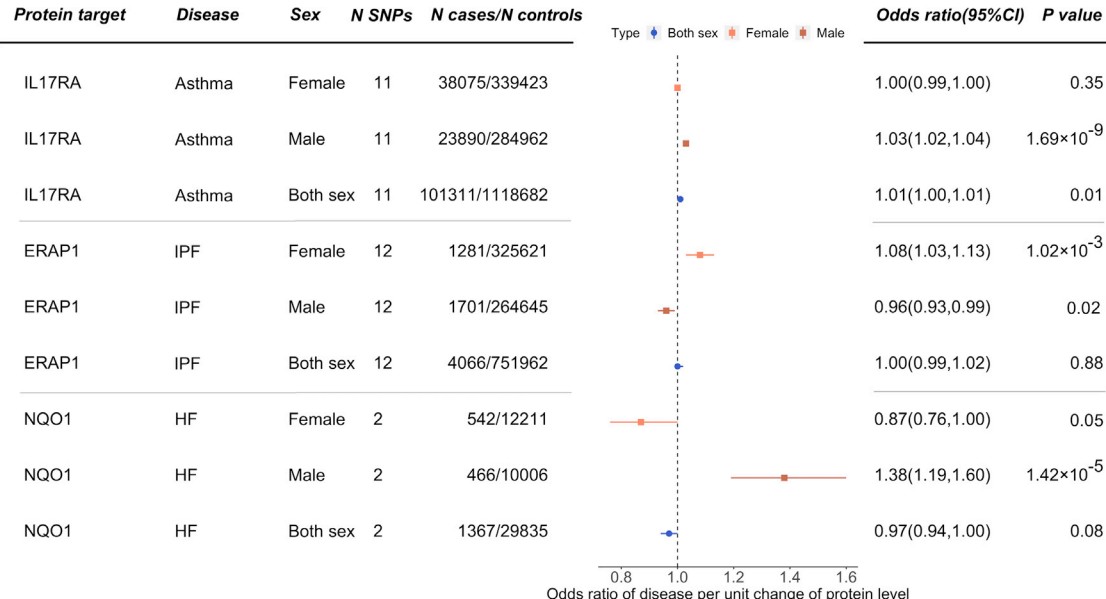

**Figure 2. Proteome MR signals showed distinguished effects in males and females**
The sex-combined and sex-specific MR estimates were presented for each protein-disease pair.

proteins into four possible situations based on the pQTL signals across ancestries (Figure 3A): (1) a protein within a genomic region that has pQTL signals in both ancestries but for which the signals were not shared across ancestries; (2) a protein within a genomic region that has one or more shared pQTLs across ancestries (and without non-shared pQTLs); (3) a protein within a genomic region that has both shared and non-shared pQTLs; and (4) a protein within a region that has pQTLs in only one of the ancestries. As shown in Figures 3B and 3C, among the tested proteins, 1,076 of them are in regions with pQTLs in both ancestries (situation 1, 2, or 3). The remainder comprise 234 proteins within regions with European-specific pQTLs and 235 proteins within regions with African-specific pQTLs (situation 4; Tables S13A and S13B). For the 1,076 proteins within regions with multiple pQTL signals, 974 had shared pQTLs in the relevant regions (situation 2 or 3; Figure 3C; Table S14). The remaining 102 proteins had only non-shared pQTLs in the regions (situation 1; Figure 3C; Tables S15A and S15B).

### Identification of multi-ancestry and ancestry-specific MR signals
We conducted a multi-ancestry comparison for the 59 unique protein-disease pairs that showed robust MR evidence (FDR < 0.05) in at least one ancestry (where ABO effect on VTE appeared in both ancestries; Tables S7A and S8A). Using an FDR threshold of 0.05 based on the 59 protein-disease pairs, we identified 7 pairs with MR signals in both ancestries (Figure 4A; Table S16A). Comparing the MR effect estimates of these seven protein-disease pairs, we observed a very high correlation of the MR effect estimates across ancestries (Pearson correlation = 85.8%; Table 1; Figure 4B). After applying a comprehensive ancestry-specific comparison, we identified seven European-specific and seven African-specific signals with MR and

colocalization evidence, while two signals showed robust MR evidence in both ancestries (Figures 4C and 4D).

In more detail, we first considered the colocalization evidence from the discovery MR (Figure 5A), in which two protein-disease pairs showed colocalization evidence in both ancestries. These include genetically predicted protein level of SERPINE2 associated with VTE (OR in Europeans = 0.94, 95% CI = 0.92–0.96, p = 7.3 × 10$^{-8}$, colocalization probability = 99%; OR in Africans = 0.82, 95% CI = 0.67–0.95, p = 1.28 × 10$^{-2}$, colocalization probability = 100%; Figure 5B) as well as the above-mentioned genetically predicted protein level of ABO associated with VTE (Figure 5C). Further comparing the two protein-disease pairs with the validation MR signals, both of them showed MR evidence (FDR < 0.05 in validation) with the same direction of effect in discovery and validation analyses (Table S16A).

We further identified the ancestry-specific protein-disease pairs that showed MR signal in only one of the ancestries. For African MR results, we selected the 86 protein-disease pairs that showed marginal MR signals (p < 0.05) and colocalization evidence in the discovery MR as candidates (Tables S8A and S8B) and applied an FDR threshold of 0.05 based on the 86 pairs. After filtering, 14 pairs passed the FDR threshold. Two of the 14 pairs overlapped with the multi-ancestry MR list and were excluded from the African-specific signal list. Among the remaining 12 pairs (Tables 1 and S16B), 7 of them showed marginal MR signals and LD check evidence in the African validation MR, which included a genetically predicted effect of protein level of SERPINF1 on stroke, angiotensin-converting enzyme (ACE) level on COPD, B4GALT6 level on POAG, F7 level on stroke, LY75 level on asthma, allograft inflammatory factor 1 (AIF1) level on HF, and CD248 level on gout (Table S16B).

For the European MR results, we selected 341 protein-disease pairs with marginal MR evidence (p < 0.05) and

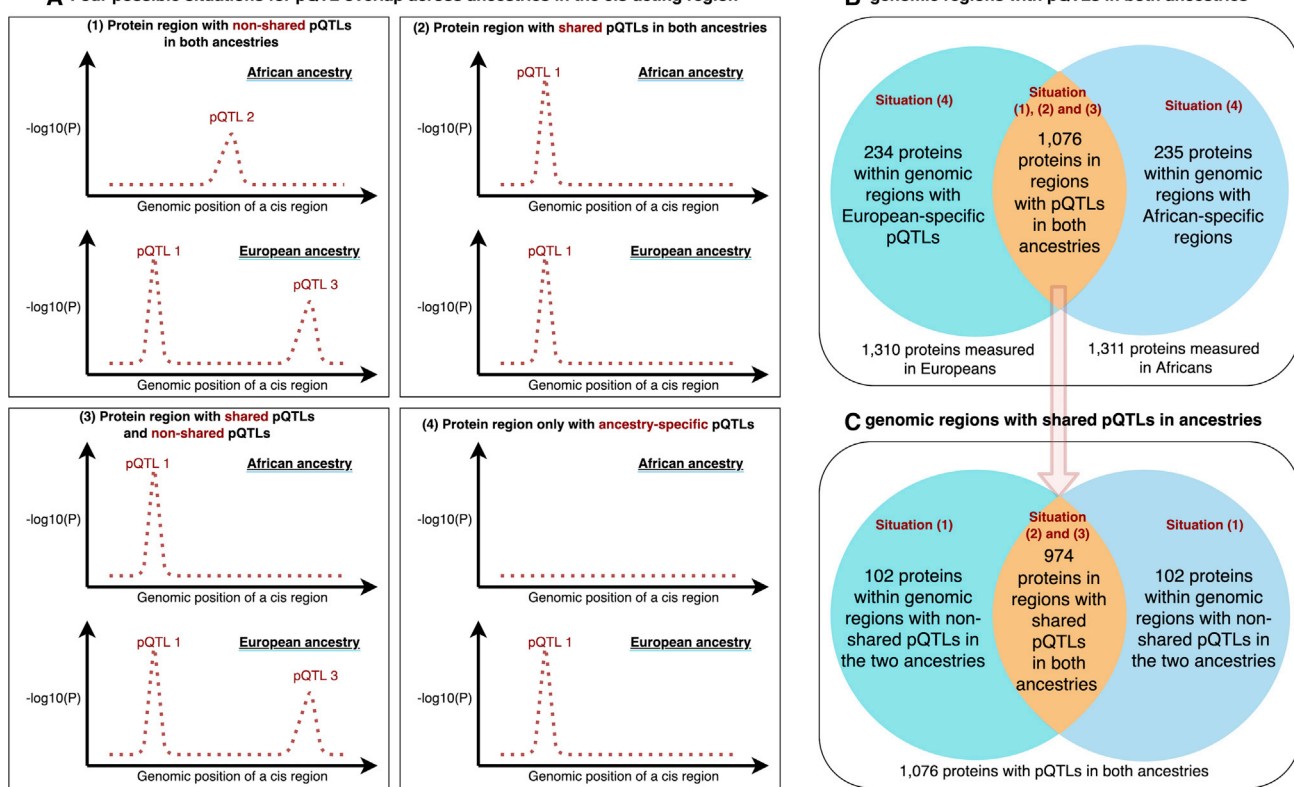

**Figure 3. Multi-ancestry investigation identifying shared protein regions or shared pQTLs**

(A) Four situations to identify multi-ancestry and ancestry-specific protein regions: (1) protein regions only non-shared pQTLs across ancestries, (2) protein regions with one or more shared pQTLs across ancestries (and no non-shared pQTLs), (3) protein regions with both shared and non-shared pQTLs, and (4) protein regions with pQTL in only one of the ancestries.

(B) Number of protein regions with ancestry-specific pQTLs.

(C) Number of protein regions with shared pQTLs in the *cis* region.

colocalization evidence in the discovery analysis as candidates. After filtering by FDR threshold of 0.05 based on the 341 pairs, 95 pairs remained (Tables 1 and S16C). Six of these overlapped with the multi-ancestry MR list and were excluded from the European-specific signal list. Further filtering the remaining 89 signals based on the validation MR evidence, 7 of them showed MR and LD check evidence in the European validation analysis, including effect of F11 level on VTE, KLKB1 level on VTE, ERAP1 level on POAG, tumor necrosis factor superfamily member 12 (TNFSF12) level on asthma, extracellular matrix protein 1 (ECM1) level on IPF, CD109 level on VTE, and IL-7R level on asthma (Table S16C).

**Drug target prioritization using MR, observational, and clinical trial evidence**

Triangulation of evidence from genetics, observational study, and clinical trials has the potential to increase the reliability of putative causal inference.[24,40] The 16 protein-disease pairs with MR and colocalization evidence in both discovery and validation were selected as candidates for this analysis (Figure 4C). We conducted observational analysis in up to 3,172 participants from the Trøndelag Health Study (HUNT)[41] (details are in STAR Methods). The logistic regression of proteins

on diseases suggested that 9 of the 16 observational associations showed the same direction of effects as the MR signals. Three of the 16 observational associations passed FDR < 0.05 in this analysis, which included ACE level on COPD, AIF1 level on HF, and SERPINF1 on stroke (Table S17).

We further mined clinical trial evidence for the 16 prioritized protein-disease pairs using the OpenTargets[42] and DrugBank[43] database. As summarized in Table 2, we found clinical trial evidence (phase 4 trials versus placebo; ClinicalTrials.gov: NCT01014338) to support our MR signal of protein level of ACE on COPD, in which the evidence was obtained from Europeans. Our study validated the efficacy of ACE inhibition on reducing COPD in African ancestry (OR in Africans = 0.88, 95% CI = 0.81–0.95, p = 1.64 × 10$^{-3}$; Table S16). Additionally, we also observed seven proteins that are drug targets of existing drugs, in which our MR signals indicate potential drug-repurposing opportunities of these drug targets to other indications (Table 2). For example, KLKB1 protein is the target for ecallantide, which is used to treat hereditary angioedema. Our study showed strong genetic evidence to support the putative causal role of inhibition of protein level of KLKB1 on reducing VTE risk (OR = 0.78; p = 4.59 × 10$^{-15}$), which implies

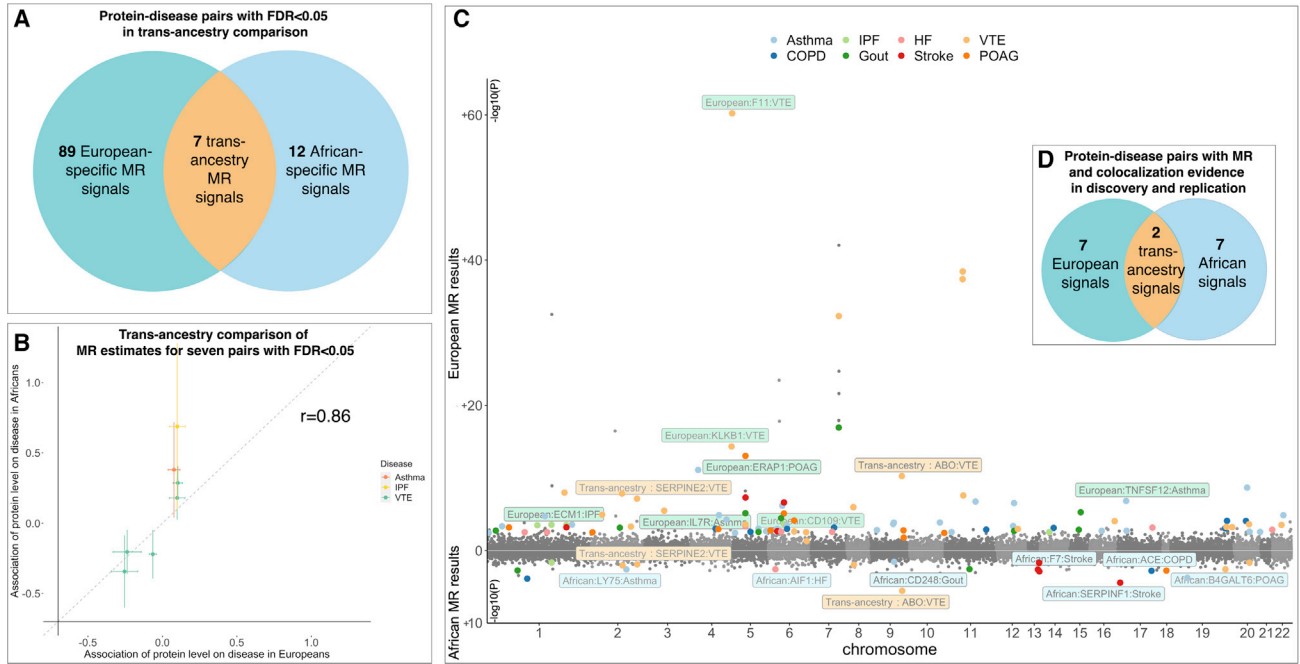

**Figure 4. Comparison of multi-ancestry proteome MR signals in European and African ancestries**

(A) Protein-disease pairs with FDR < 0.05 in multi-ancestry comparison.

(B) Comparison of MR effect estimates of the seven protein-disease pairs with MR evidence (FDR < 0.05 in multi-ancestry analysis); each point refers to one protein-disease pair, the x axis refers to the MR estimate in European ancestry, and the y axis is the MR estimate in African ancestry.

(C) Miami plot of the protein-disease putative causal estimates in European and African ancestries; each point refers to a protein-disease pair, the x axis is the chromosome and position of the protein, the y axis is the −log10(p) of the MR estimate in European (upper) and African ancestry (bottom); the points with colors refer to the 7, 12, and 89 protein-disease pairs with multi-ancestry, African-specific, or European-specific MR effects (FDR < 0.05 in multi-ancestry analysis); different color refers to different outcomes of the protein-disease pairs; the points with legends are the two, seven, and seven protein-disease pairs that showed MR and colocalization evidence in discovery and validation MR analyses; background colors in the legends refer to multi-ancestry (yellow), European-specific (green), and African-specific (blue) MR estimates.

(D) Protein-disease pairs with MR (FDR < 0.05) and colocalization evidence in multi-ancestry comparison and validation analysis.

a repurposing opportunity of ecallantide on VTE prevention. In addition, the effect of ABO level on VTE was also observed in recent genetic studies, including the VTE GWAS meta-analysis conducted in the GBMI consortium.[44] The remaining seven protein-disease pairs we identified were considered as new putative causal proteins and are therefore potential new drug targets (Table S18). For example, the serpin-related protein, SERPINE2, has previously been reported to be associated with COPD.[45] This association was confirmed by our MR results (MR p = $9.28 \times 10^{-5}$). Our disease-wide scan further suggested its effect on VTE (p = $7.3 \times 10^{-8}$) and IPF (p = $9.91 \times 10^{-3}$). This implies that SERPINE2 could be considered an attractive drug target for prevention of COPD, IPF, and thromboembolism (Figure S3).

To prioritize the most valuable drug targets from this study, we summarized the drug target prioritization analyses we conducted, which included four filtering steps: discovery MR, sensitivity analyses and validation MR, multi-ancestry comparison, and triangulation. As shown in Figure 6A, eight protein-disease pairs were ranked as the most valuable findings after the filtering, which include IL-7R and TNFSF12 level on asthma; ACE level on COPD; AIF1 level on HF; SERPINF1 level on stroke; and SERPINE2, F11, KLKB1, and ABO level on

VTE (Figure 6B; Table S18). Some of these pairs showed robust MR evidence, for example, SERPINE2 and ABO associated with VTE. Some of these pairs showed integrative evidence, for example, the effect of ACE on COPD was validated by genetic, observational, and clinical trial evidence. To define novelty, we further compared the eight prioritized protein-disease pairs with existing pQTL GWAS and MR findings reported in 10 proteome studies.[5–9,19,21,46–48] Two of the pairs, SERPINF1 on asthma and KLKB1 on VTE, were not reported in any of the pQTL studies and are not under clinical investigation yet, which we have therefore suggested are completely novel protein-disease pairs identified in this study. Five pairs were reported in previous pQTL studies but are not under clinical investigation yet, which we have suggested as novel drug targets. The pair ACE on COPD was not reported in any pQTL studies but is under a phase 4 trial; therefore, we considered this to be an existing drug/target with potential to be validated in African ancestry (Figure 6B; Table S18). Our study provides evidence to support formal investigations of these protein-disease pairs in future clinical trials.

It is important to note that our drug target prioritization was mainly based on the use of a p value threshold for MR

**Table 2. Drug target validation and repurposing opportunities, related to STAR Methods**

| Protein-disease MR information | | | DrugBank information | | | | OpenTargets information | | | |
|---|---|---|---|---|---|---|---|---|---|---|
| Protein | Gene | Outcome | DrugBank ID | Drug name | Molecular action | Current clinical indication(s) | Overall score | Genetic score | MR evidence from this study | Evidence synthesis |
| Angiotensin-converting enzyme | ACE | chronic obstructive pulmonary disease | DB00492 | fosinopril | inhibitor | hypertension, chronic obstructive pulmonary disease | 0.4 | 0 | $1.64 \times 10^{-3}$ | validating efficacy in African ancestry |
| Coagulation factor VII | F7 | stroke | DB00036 | coagulation factor VIIa recombinant human | inhibitor | hemophilia A and B | 0 | 0 | $2.42 \times 10^{-3}$ | repurposing |
| Allograft inflammatory factor 1 | AIF1 | heart failure | DB03147 | flavin adenine dinucleotide | unknown | vitamin $B_2$ deficiency | 0.03 | 0 | $2.67 \times 10^{-3}$ | repurposing |
| Coagulation factor XI | F11 | venous thromboembolism | DB00100 | coagulation factor XI | ligand | hemophilia B, hereditary angioedema | 0.49 | 0.79 | $6.14 \times 10^{-61}$ | repurposing |
| Kallikrein B1 | KLKB1 | venous thromboembolism | DB05311 | ecallantide | inhibitor | hereditary angioedema | 0.05 | 0.08 | $4.59 \times 10^{-15}$ | repurposing |
| ER Aminopeptidase 1 | ERAP1 | primary open-angle glaucoma | DB11781 | tosedostat | inhibitor | acute myeloid leukemia, pancreatic cancer, and multiple myeloma | 0 | 0 | $9.19 \times 10^{-14}$ | repurposing |
| Tumor Necrosis Factor (TNF) Superfamily Member 12 | TNFSF12 | asthma | N/A | BIIB-023* | inhibitor | lupus nephritis | 0.01 | 0 | $1.50 \times 10^{-7}$ | repurposing |
| Interleukin-7 receptor | IL7R | asthma | DB08895 | tofacitinib | inhibitor | rheumatoid and psoriatic arthritis | 0.5 | 0.79 | $3.71 \times 10^{-3}$ | repurposing |

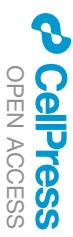

**Figure 5. Regional genomic plots of two protein-disease pairs with MR and colocalization evidence of potential causality in European and African ancestries**

(A) The two theoretical models related to genetic colocalization, causality, and colocalized, as well as confounding by LD.

(B) Regional plots of protein level of SERPINE2 on VTE in European and African ancestries.

(C) Regional plots of protein level of ABO on VTE in European and African ancestries.

evidence (FDR < 0.05 based on gIVW estimates), which we simply use as a heuristic for highlighting putative causal effects worthy of follow-up. Investigations of results can therefore apply more (or less) stringent thresholds by filtering the protein-disease effects downloadable from our web browser (https://epigraphdb.org/multi-ancestry-pwmr/).

## DISCUSSION

The application of GWASs to investigate complex traits and diseases is now over 15 years old.[49] With increasing participation in major consortia such as GBMI, we are now entering a new era of multi-ancestry meta-analysis of GWASs across biobanks, which

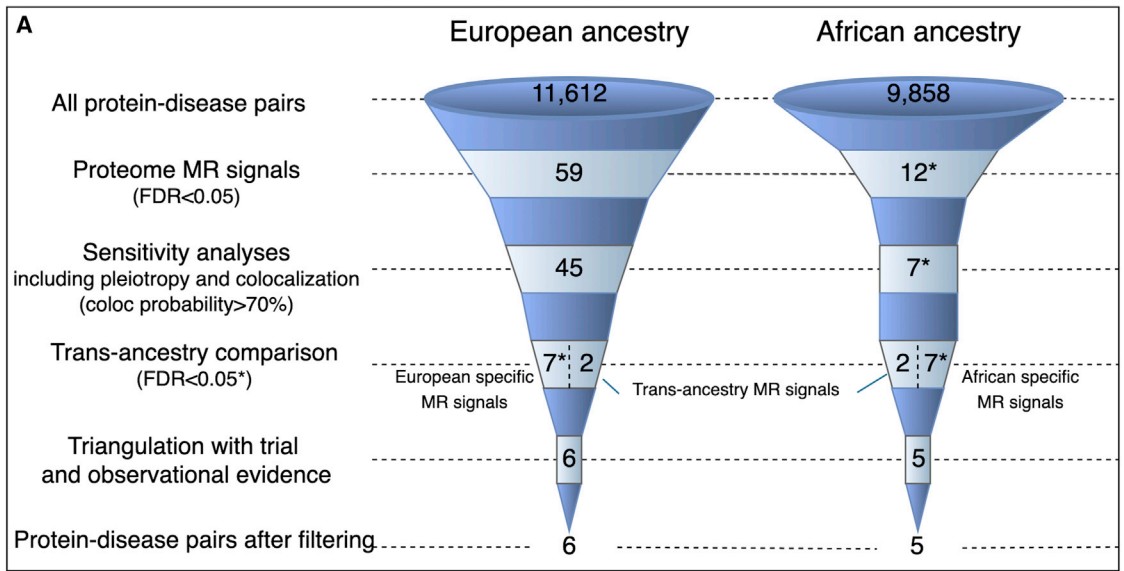

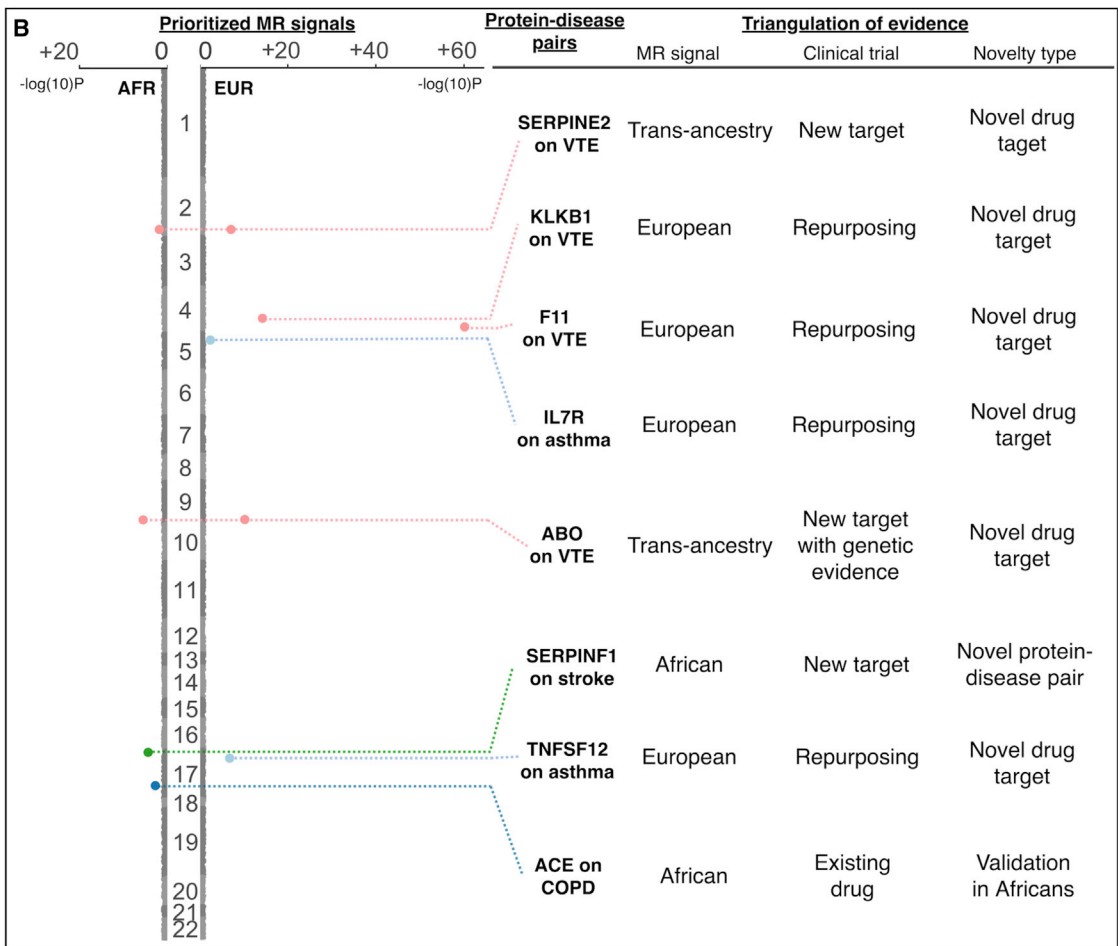

**Figure 6. Drug target prioritization profiles of MR signals across European and African ancestries**
(A) Drug target prioritization profile of this study.
(B) Evidence level for the eight prioritized drug targets (details are in Tables S17 and S18).

provides new opportunities and challenges. In this study, we utilized the multi-ancestry GWAS data from GBMI to implement a proteome-wide MR in two ancestries. By estimating 21,470 putative causal effects of proteins on preventing relevant diseases in European and African ancestries, we found evidence for potential causal effects in 45 and 7 protein-disease pairs in the two ancestries, respectively, with both MR and colocalization evidence. After formal comparison of MR estimates across ancestries, two protein-disease pairs showed shared putative causal effects across the two ancestries, seven showed European-specific putative causal effects, and seven showed African-specific effects. All the mentioned comparison results showed MR and colocalization evidence in both multi-ancestry and validation analyses. By triangulating these 16 putative causal MR signals with clinical trials and observational evidence, we validated the efficacy of ACE inhibition on COPD and generalized its effect to those of African ancestry, suggested seven drug-repurposing opportunities, and identified seven novel protein-disease pairs that warrant further investigation, for example, the effects of SERPINE2 on VTE. Collectively, we highlighted the value of multi-ancestry MR as an approach to inform the generalizability of drug target efficacy across ancestries using GWAS results from the GBMI.

A major issue for generalizability of drug targets across populations is that most clinical trials are carried out in European ancestry and work under the assumption that drug effects are consistent across ancestries, which is not always the case.[50] We have shown that multi-ancestry proteome MR offers the opportunity to address this bias by enabling us to estimate the effects of drug targets in different populations, which could be useful evidence to support the design of multi-ancestry clinical trials. In this study, we identified multi-ancestry putative causal effects of two protein-disease pairs, despite being limited by statistical power. This highlights the importance of large-scale genetic studies in different populations, which should be a key priority for the research community.

In addition to generalizing targets across ancestries, our MR approach also identified evidence of potential heterogeneity of drug response between ancestries for 14 protein-disease pairs. For example, protein levels of SERPINF1 showed MR evidence to support a putative causal effect on stroke in African ancestry ($p = 3.76 \times 10^{-5}$) but showed little evidence of an effect in European ancestry ($p = 0.83$) (see Table S16B, the full results could be queried using the EpiGraphDB web application). Moreover, some recent trans-ancestry fine mapping studies have focused on estimating the influence of potential pleiotropy on the putative causal variant identification by using multi-ancestry GWAS datasets.[15,16] Because of the limited power of the African datasets, it is still challenging to clearly claim heterogeneity of drug response and test for pleiotropy in this study, but we hope future studies will address these questions as larger datasets become available. Our multi-ancestry MR pipeline (including multi-ancestry application of PWCoCo) and pQTL comparison approach implemented in TAMR provide a useful framework for such future studies.

The concept of generalizability of drug target effects could be extended to identify sex-specific effects. In this study, we identified 15 MR signals with robust evidence to support sex-specific effects. In addition to the protective effect of IL-17RA inhibition on asthma in males, our results further suggested reconsidera-

tion of efficacy of two protein-disease pairs (see Table S11). ERAP1 is the protein target of an anti-leukemia drug, tosedostat, and NQO1 is the protein target of an anti-neurodegressive drug, vatiquinone. However, our study found that the estimated putative causal effects of ERAP1 level on IPF and NQO1 level on HF were relatively strong, but the effect estimates were in opposite directions in males and females. Whether these targets and/or drugs may have different drug responses in males and females needs further investigation. In the future, more comprehensive sex-specific proteome MR could be conducted using sex-stratified pQTLs against male- and/or female-only diseases, e.g., on pregnancy and perinatal outcomes[51] to predict drug target effects in pregnant women.

Our study also provides methodological guidance for future proteome MR. Previously, we showed that naive application of MR without sensitivity analyses may yield over 30% unreliable results.[6] A recent study further suggested that 51% of results from transcriptome-wide association studies could not be confirmed by genetic colocalization.[52] Another study showed the importance of distinguishing disease-causing gene expression from disease-induced gene expression by evaluating reverse causality using genetic data.[28,53] In this study, we considered these alternative explanations (including reverse causality, confounding by LD, and horizontal pleiotropy) and developed a pipeline that integrates some novel sensitivity analyses (e.g., TAMR for multi-ancestry pQTL comparison and extension of PWCoCo in multiple ancestries). In the future, integrating our proteome MR pipeline with other methods, including transcriptome-wide association study[54] and drug discovery pipeline,[36] will provide more robust evidence to support causal gene/protein identification and drug target prioritization.

### Limitations of the study

Our study has several limitations. First, the statistical power of the African-specific pQTL data and disease GWASs were still limited compared with the European datasets. Although GBMI has incorporated one of the largest consortia of GWASs in African ancestry, the number of MR results reaching our evidence threshold in African ancestry was still 6.4 times lower than in European ancestry. This is mainly because the sample size of the African pQTLs was 3.8 times lower than that of the European pQTLs (7,213 European ancestry versus 1,871 African ancestry), and some of the disease, e.g., IPF, has a limited number of cases in African ancestry (see Table S6A). Second, to reduce the possibility of identifying false-positive findings, we applied the FDR threshold in discovery, validation, and multi-ancestry comparison separately. We further applied a rigorous set of sensitivity analyses and triangulated the MR findings with observational and clinical trial evidence. This increased the reliability of our top findings; however, some of them still showed relatively wide CIs. We therefore recommend caution in interpreting the results and further validation of findings in future proteome MR studies. Third, given the increased power of the pQTL study, 79.1% of the proteins now have two or more conditional independent pQTL signals in the *cis* regions. We therefore applied some conventional MR sensitivity approaches, including the MR-Egger approach, which can test for horizontal pleiotropy. However, we note that use of MR-Egger with few instruments may be biased because

it is unlikely that the InSIDE assumption will be satisfied because the sample correlation between the pleiotropic effects and instrument strengths could be quite large by chance. Ideally, over 30 variants are needed to ensure that the MR-Egger bias term settles close to zero.[33] Fourth, the disease cases were defined using Phecodes such as 10th revision of the International Statistical Classification of Diseases and Related Health Problems (ICD10) codes. There is a potential limitation that the quality of definition of the disease cases may not be as reliable as clinically diagnosed cases. According to the comparison conducted in the GBMI flagship and disease-specific GWAS papers, the genetic effects for clinically diagnosed IPF,[86] VTE,[44] and AAA[23] were similar compared with those for the Phecode-defined data. However, the genetic effect of asthma showed quite different effect estimates using the two definitions.[87] Fifth, for the validation analysis in European ancestry, the pQTLs were obtained from studies using SOMAscan v.3 platform, while the discovery pQTLs were obtained from studies using SOMAscan v.4 platforms. We observed some levels of departure for pQTL effect sizes across these studies (see Table S4), but whether this is due to version difference of SOMAscan or other differences needs further investigation. Finally, because we included non-specific pQTLs as instruments in this study, some of the MR findings could be influenced by pleiotropy. For instance, pQTLs within the ABO region are known to be pleiotropic and associated with multiple proteins.[5] In this study, we identified the putative causal role of ABO level on VTE, which aligns well with existing and novel genetic evidence of this protein-disease pair.[44] This example demonstrated that proteome MR results using pleiotropic instruments may still yield additional evidence to support drug target prioritization, although caution and extra validation (e.g., pleiotropy test) are needed when interpreting the results.

### Recommendations for multi-ancestry MR in the era of global biobank meta-analysis

Although proteome-wide MR shows promise in drug target prioritization, there is little consistency in analytical strategy and the approach to reporting MR findings. Recently, the Strengthening the Reporting of Observational Studies in Epidemiology using Mendelian Randomization (STROBE-MR) statement has been released to define the reporting standard of MR findings (https://www.strobe-mr.org/).[55,56] To complement these guidelines, we list here some specific challenges of MR and provide some specific recommendations for multi-ancestry proteome-wide MR in the era of global biobank meta-analysis.

### Selection and validation of genetic instruments for proteins

Instrument selection is a key step for all types of MR analysis. The selection of instruments for proteome MR has been discussed previously.[3,57,58] Our study provides a comprehensive pipeline for pQTL instrument selection and validation. We also provide some clues for three key challenges:

1. biological way to categorize pQTLs is into *cis*-acting pQTLs (which are pQTLs that are within or close to the protein-coding gene) and *trans*-acting pQTLs (which are pQTLs that are not within or close to the protein-coding gene). In general, we recommend using *cis*-acting pQTLs

as genetic instruments for future MR because they are considered to have a higher prior probability of specific biological effects.[57,59] However, *cis*-pQTLs that alter epitope binding sites can lead to potential false-negative variant-protein association signals, resulting in biased MR estimates,[3,5] which need to be used with caution. Where *cis*-acting variants are not available, non-pleiotropic *trans*-acting pQTLs could be considered as a backup if they can map to functioning genes.[7,60]

2. Most proteome GWASs[5,8,9,19,61,46,47] and MR[6,7,48] studies have been conducted in plasma samples, and there is little tissue-specific data for the human proteome. Recently, some studies have identified brain- and cerebrospinal fluid-specific pQTLs and further identified their roles on brain-related diseases using MR.[62–64] This type of study may be able to detect tissue-specific protein effects on diseases.

3. Winner's curse of GWAS estimates will bias the downstream MR estimates toward the null when exposure and outcome samples are independent. A recent study suggested that winner's curse incurs substantial overestimation of effect sizes in a mean of 35% of discovered associations per trait in UK Biobank.[65] The same study suggested that a three-sample MR setting[66] using two independent exposure datasets may reduce the influence of winner's curse. In addition, some recent studies have suggested that the use of genome-wide instruments and inclusion of weak instruments may improve the reliability of MR estimates and increase power compared with a classic "significant SNP" approach.[65,67,68]

### Selection of outcomes

The outcome selection is the other key step in proteome MR. We summarize four key considerations:

1. valuating drug efficacy on progression of a disease using MR in disease cases will represent treatment of disease.[69] However, until now, most of the MR studies have been conducted using disease incidence data in a case-control setting rather than in disease progression. The future development of global biobank meta-analysis is likely to create a valuable source of data for studying disease progression. One advantage of studying disease progression in biobanks is that many biobanks have linked participants with their electronic health record, making it easier to obtain disease progression information. In addition, some novel genetic epidemiology methods, such as the Slope-Hunter approach,[70] have been developed to detect and adjust potential selection bias introduced by disease progression data.[71]

2. Most MR analyses also assume that genetic effects on proteins are consistent in different subgroups of the population (e.g., in males and females; in diseased patients and healthy control subjects). However, naive use of genetic effects generated in a general population as a proxy in disease subgroups may yield biased estimates of putative causal relationships, which we have illustrated recently for C-reactive protein.[72] GBMI provides ancestry-specific and sex-specific GWAS data and clearly defines

disease cases and controls. Both features offer the opportunity to implement MR in subgroups of the populations in the future.

3. For "two-sample" genetic approaches (such as genetic correlation, polygenic risk score association, transcriptome-wide association study, and MR), a less considered issue is the consistency of covariates that were used in the exposure (i.e., protein) GWAS and outcome (i.e., disease) GWAS. A recent study suggested that this can produce biased two-sample MR estimates.[73] Future GWASs of GBMI and related biobanks could consider providing multiple GWASs for each trait, with differing sets of covariates and/or environmental factors (e.g., with and without BMI).

4. In comparisons of MR and RCT, the exposure and outcome of the two approaches need to be as close as possible, for example, the outcome definition needs to be similar (e.g., similar population disease status, age, and sex).

### Proteome-wide MR and sensitivity analyses

A large collection of MR and sensitivity methods can be used to estimate the putative causal roles of proteins on diseases. We have listed a few general recommendations here:

1. For discovery analysis, Wald ratio and IVW work effectively.[29,74] However, to increase power and reliability of the MR estimate, generalized IVW and MR-robust adjusted profile score (MR-RAPS) are potential alternatives.[30,75]

2. For sensitivity analyses, given the limited number of *cis*-acting pQTLs for each protein, it is difficult to apply classic sensitivity methods, such as weighted median and mode estimator methods to test for MR assumptions. The following methods are recommended for current and future proteomics MR:

   a. Genetic colocalization is important in distinguishing causality from confounding by LD.[35,76,77] Such confounding could cause a false inference of a putative causal effect of the drug target on the disease. Some recent methods have been developed to relax the single putative causal variant assumption in colocalization.[6,78,79]

   b. MR Steiger filtering is a method that was designed to estimate the directionality of exposure-outcome effects,[27] which is key to addressing potential reverse causality.

   c. Some recent methods such as causal analysis using summary effect estimates (CAUSE) account for correlated and uncorrelated pleiotropy using genome-wide summary statistics[80] but are not optimized for analysis in a specific genomic region. Other methods such as constrained maximum likelihood and model averaging (cML-MA),[81] genome-wide MR analysis under pervasive pleiotropy (GRAPPLE),[82] and MA-APSS[83] were developed to deal with correlated and uncorrelated pleiotropy. Based on our pilot comparison, cML-MA provided the best statistical power and accuracy among these methods in a drug target MR setting (see Table S19). With more efforts in methods development and evaluation of other methods such as contamination mixture method[84] and MR locus,[85] these approaches may effectively estimate the level and impact of pleiotropy on causal estimates in the near future.

3. In addition to novel target identification and drug repurposing, MR can also inform on-target safety events of drug targets, potentially addressing some of the approximately 30% of new drug discovery programs that fail because of safety considerations.

### Conclusions

In summary, this MR study systematically investigated protein effects on eight complex diseases across European and African ancestries, providing valuable evidence to inform the generalizability of drug targets to other less studied ancestries. We anticipate that a new era of proteome MR will soon emerge, using new proteome resources from large-scale biobanks such as UK Biobank and CHARGE. Our findings, analysis pipeline, and recommendations on proteome MR will help future studies design, conduct, and interpret multi-ancestry proteome-wide MR.

### STAR★METHODS

Detailed methods are provided in the online version of this paper and include the following:

- KEY RESOURCES TABLE
- RESOURCE AVAILABILITY
  - Lead contact
  - Materials availability
  - Data and code availability
- EXPERIMENTAL MODEL AND SUBJECT DETAILS
  - Atherosclerosis Risk in Communities study (ARIC)
  - African American study of Kidney Disease and Hypertension Cohort study (AASK)
  - Interval
  - Global Biobank Meta-analysis Initiative (GBMI)
  - The Trøndelag Health study (HUNT)
- METHOD DETAILS
  - Genetic instrument selection of plasma proteome
  - Genetic instrument validation
  - Validation of instruments using directionality test
  - Validation of instruments using heterogeneity test and specificity test of instruments
  - Outcome selection in the global biobank meta-analysis initiative
  - Discovery proteome-wide MR of complex diseases in European and African ancestries
  - Sensitivity analysis of candidate MR signals
  - Estimation of horizontal pleiotropy and heterogeneity of MR signals
  - Genetic colocalization analysis of the candidate MR signals
  - Estimation of potential aptamer binding artificial effect of pQTLs
  - Validation MR analysis
  - Sex-specific MR of candidate protein-disease signals
  - Multi-ethnic comparison of pQTL and proteome MR effect estimates across ancestries
  - Estimation of multi-ancestry and ancestry-specific pQTLs

○ Estimation of multi-ancestry and ancestry-specific proteome MR signals
○ Triangulation of protein-disease MR signals with observational and clinical trial evidence
○ Drug target prioritization profiling

## SUPPLEMENTAL INFORMATION

## ACKNOWLEDGMENTS

The authors thank Dr. Shinichi Namba for the internal GBMI review of this manuscript. J.Z. was supported by the Academy of Medical Sciences (AMS) Springboard Award; the Wellcome Trust; the Government Department of Business, Energy and Industrial Strategy (BEIS); the British Heart Foundation and Diabetes UK (SBF006\1117); and the Vice-Chancellor Fellowship from the University of Bristol. G.D.S., T.R.G., G.H., J.Z., H.R., Y.C., and Y.L. work in a unit supported by the Medical Research Council for the Integrative Epidemiology Unit (MC_UU_00011/1 & 4) at the University of Bristol. T.R.G. holds a Turing Fellowship from the Alan Turing Institute. L.B. works in a research unit supported by the K.G. Jebsen Center for Genetic Epidemiology supported by Stiftelsen Kristian Gerhard Jebsen; Faculty of Medicine and Health Sciences, NTNU; The Liaison Committee for education, research and innovation in Central Norway; and the Joint Research Committee between St. Olavs Hospital and the Faculty of Medicine and Health Sciences, NTNU. The UK Medical Research Council and Wellcome (grant 102215/2/13/2) and the University of Bristol provide core support for ALSPAC. This work was supported by Cancer Research UK, Integrative Cancer Epidemiology Programme (grant C18281/A29019) and Elizabeth Blackwell Institute for Health Research, University of Bristol. A comprehensive list of grant support is available on the ALSPAC website (http://www.bristol.ac.uk/alspac/external/documents/grant-acknowledgements.pdf). The views expressed in this publication are those of the author(s) and are not necessarily those of the NHS, the National Institute for Health Research, or the Department of Health and Social Care. This publication is the work of the authors, and J.Z. will serve as guarantor for the contents of this paper. We gratefully acknowledge all studies and biobanks that have contributed to the GBMI: BioBank Japan, BioMe, BioVU, Canadian Partnership for Tomorrow, Colorado Center for Personalized Medicine, China Kadoorie, deCODE Genetics, East London Genes & Health, Estonian Biobank, FinnGen, Generation Scotland, HUNT, Lifelines, Michigan Genomics Initiative, Million Veteran Program, Netherlands twin register, Partners Biobank, QIMR Berghofer–QIMR Biobank (QSkin and GenEpi), Taiwan Biobank, UCLA Precision Health Biobank, and UK Biobank. We gratefully acknowledge Zhang et al., who made the proteome QTL data publicly available (http://nilanjanchatterjeelab.org/pwas/).

## AUTHOR CONTRIBUTIONS

J.Z., T.R.G., B.M.N., W.Z., and B.M.B. conceived and designed the study and oversaw all analyses. H.Z. performed the MR analysis. H.Z. conducted the sensitivity analysis, including colocalization analysis. H.R., T.H.N., L.B., and B.M.B. conducted the observational analysis using HUNT data. G.H., Y.C., and J.Z. performed the pQTL and MR results comparison. J.Z. performed the triangulation and drug target prioritization analysis. Y.L. developed the database and web browser. H.Z. and J.Z. wrote the manuscript. G.D.S., B.M.B., W.Z., B.M.N., and T.R.G. reviewed the paper and provided key comments.

## DECLARATION OF INTERESTS

J.Z., T.R.G., and G.D.S. received funding from Biogen for other work on drug target prioritization. B.M.N. is on the scientific advisory board at Deep Genomics and Neumora and is a consultant for Camp4 Therapeutics, Takeda Pharmaceutical, and Biogen.

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

## Article

CellPress

## STAR★METHODS

### KEY RESOURCES TABLE

| Reagent or resource | Source | Identifier |
| --- | --- | --- |
| GWASs of plasma proteome | Zhang et al. 2021[21] | https://www.biorxiv.org/content/10.1101/2021.03.15.435533v1.full |
|  | Zheng et al. 2020[6] | https://www.nature.com/articles/s41588-020-0682-6 |
| GWASs of diseases | GBMI – Zhou et al. 2021[23] | https://www.globalbiobankmeta.org/ |
| Software and algorithms | | |
| MR models | Hemani et al. 2018[88] | https://github.com/MRCIEU/TwoSampleMR |
|  | Yavorska et al. 2017[89] | https://github.com/cran/MendelianRandomization |
| Colocalization analysis | Giambartolomei et al. 2013[35] | https://github.com/chr1swallace/coloc |
| Pairwise conditional and colocalization analysis | Robinson et al. 2021[90] | https://github.com/jwr-git/pwcoco/ |
| EpiGraphDB multi-ancestry proteome MR browser | This study | https://epigraphdb.org/multi-ancestry-pwmr |
| Multi-ancestry pQTL comparison | This study | https://github.com/universe77/TAMR |
| Other | | |
| Proteome MR pipeline | This study | https://github.com/globalbiobankmeta/multi-ancestry-pwmr |

## RESOURCE AVAILABILITY

### Lead contact
The multi-ancestry proteome Mendelian randomization pipeline was shared via the GBMI GitHub repository (https://github.com/globalbiobankmeta/multi-ancestry-pwmr) and Zenodo (https://doi.org/10.5281/zenodo.7087319). Further information and requests for resources and reagents should be directed to and will be fulfilled by the lead contact, Jie Zheng (jie.zheng@bristol.ac.uk).

### Materials availability
This study did not involve any other unique materials.

### Data and code availability
The pQTL GWAS summary statistics used in the paper are freely downloaded from ARIC website (http://nilanjanchatterjeelab.org/pwas/). You could also access it on Zenodo with https://doi.org/10.5281/zenodo.7087319 (https://zenodo.org/record/7087319#.YyTgFpMtWS4). The GBMI GWAS summary statistics used are freely accessible on the GBMI website (https://www.globalbiobankmeta.org/). All our MR estimates and colocalization results (including 11,612 protein-disease signals in European ancestry and 9,858 signals in African ancestry) are freely available to browse, query and download via the EpiGraphDB platform[1] (https://epigraphdb.org/multi-ancestry-pwmr/). An application programming interface (API) documented on the site enables users to programmatically access data from the database. An unchanging version of the code at time of publication is available at GitHub repository (https://github.com/globalbiobankmeta/multi-ancestry-pwmr) and Zenodo with https://doi.org/10.5281/zenodo.7087319.

## EXPERIMENTAL MODEL AND SUBJECT DETAILS

We accessed protein quantitative trait loci (pQTL) data from the ARIC, AASK cohorts,[21] INTERVAL study,[5] Folkersen et al[61] Yao et al.[47] and Emilsson et al.[48] The majority of the proteins were measured using the SOMAlogic platform (https://somalogic.com/). For European samples, 7,212 European ancestry individuals with protein data from the ARIC cohort were selected as discovery samples, and 3,301 Europeans with protein data from INTERVAL study and/or Emilsson et al. were selected as validation samples. For African samples, 1,871 African ancestry individuals with protein data from the ARIC cohort were selected as discovery samples, and 467 African ancestry individuals with protein data from AASK were selected as validation samples.

For the disease outcome data, we assembled GWAS data from up to 1,219,215 European samples (100,736 cases and 1,118,479 controls), and up to 32,653 African samples (5,054 cases and 27,599 controls) with genetic association information from the Global Biobank Meta-analysis Initiative (GBMI; Table S6). We further utilized individual-level protein and disease data from the HUNT study[41] to conduct an observational association analysis. Detailed information about these cohorts is listed below.

### Atherosclerosis Risk in Communities study (ARIC)
The Atherosclerosis Risk in Communities study (ARIC), sponsored by the National Heart, Lung, and Blood Institute (NHLBI) is a prospective study to investigate the etiology of atherosclerosis and its clinical side effects and variation in cardiovascular risk factors, medical care, and disease by location, gender, race and date. Starting from 1987, each ARIC field centre randomly recruited around 4,000 adults aged 45-64 years to receive extensive examinations including medical, social, and demographic data. All the subjects would be examined twice, three years apart and the follow-up is always conducted with phone calls to maintain contact and assessment. ARIC aims to use modern biochemistry and observational analysis to promote atherosclerosis study to a deeper and broader scope.[91]

### African American study of Kidney Disease and Hypertension Cohort study (AASK)
The African American Study of Kidney Disease and Hypertension Cohort Study (AASK Cohort) is an extension of the AASK Clinical Trial. It's a prospective observational study conducted in multiple centers. The primary objective of the AASK Cohort Study is to determine the possible course of renal function variation and risk factors for chronic kidney disease (CKD) progression in African-Americans with hypertensive kidney disease apart from BP control and use of recommended reno-protective, anti-hypertensive medication. A secondary objective is to determine the occurrence of cardiovascular disease and also discover and evaluate its risk factors.[92]

### Interval
The INTERVAL study is an open randomized trial comprising about 50,000 participants of varying blood donation intervals.[93] The primary goal is to determine whether blood can be safely and acceptably collected from donors by England's National Health Service Blood and Transplant (NHSBT) more frequently and at similar intervals to other European countries. The INTERVAL BioResource provides a very powerful research platform to store and analyse detailed information of donor health. Apart from Serial collection of biological samples and clinical information, the study also includes extensive genetic, haematological, biochemical lifestyle, side-effects and other donation-related characterisation of donors. All the information could be linked with electronic health records. For SomaLogic assays, Sun et al. randomly selected two non-overlapping sub-cohorts of 2,731 and 831 participants from INTERVAL. After genetic quality control, 3,301 participants (2,481 and 820 in the two sub-cohorts) remained for analysis. No statistical methods were used to determine sample size. The experiments were not randomized. Laboratory staff conducting proteomic assays were blinded to the genotypes of participants.

### Global Biobank Meta-analysis Initiative (GBMI)
Global Biobank Meta-analysis Initiative (GBMI) is a collaborative network containing multiple biobanks collaborating through meta-analysis with established resources of genotype, phenotype and GWAS to develop a global and growing resource for human genetics research (https://www.globalbiobankmeta.org/). GBMI currently represents 2.6 million research participants with health and genetic data from twenty-one biobanks across four continents. It incorporates diverse ancestries in genetic studies by including biobank samples from 6 main populations and 14 endpoints selected based on the common interest of the contributing biobanks.[23] Incorporating samples with diverse ancestries in the biobank meta-analysis enables comparison of effect sizes of genomic loci across ancestry. Also, the sex-stratified meta-analysis allows for comparing effect sizes of the genomic loci between sexes.

### The Trøndelag Health study (HUNT)
The Trøndelag Health Study (HUNT) is a population-based cohort in Trøndelag County in Norway.[41] From the third survey, HUNT3, performed in 2006–2008, protein measurements were performed in a subset of collected serum samples[94,95] and results were available in 3190 individuals for whom status for the eight disease outcomes were also available. Serum samples were analysed using the multiplexed, aptamer-based, affinity proteomics platform (SOMAscan™).

### METHOD DETAILS

### Genetic instrument selection of plasma proteome
In this study, the genetic variants associated with plasma proteins were used as genetic instruments for the MR analysis. We started the instrument selection process by accessing the pQTL data from three cohorts, ARIC, INTERVAL and AASK. We selected all conditionally independent pQTLs that were associated with proteins at a false discovery rate (FDR) < 0.05. To fit with the data requirements of the MR and colocalization analyses, we only selected pQTLs with full summary statistics available in the cis-acting regions. Although the pQTLs are conditionally independent signals, we still applied LD clumping to remove pQTLs in very strong LD with the top signals (LD $r^2$<0.6) to avoid the issue of collinearity in the MR model. For the discovery MR analysis, 6,614 pQTLs of 1,310

proteins in 7,212 Europeans from ARIC (Table S1) and 3,900 pQTLs of 1,311 proteins in 1,871 Africans from ARIC were kept after selection (Table S2).[21] Among the protein data used in the discovery analysis, 1,076 proteins were measured in both African and European ancestries and used for the multi-ancestry comparison analysis (details described in later section).

## Genetic instrument validation
### *Validation of instrument strength*
To quantify the statistical power of the pQTLs, we estimated the strength of the genetic predictors of each tested variant using F-statistics. If any pQTLs had F-statistics lower than the widely used threshold of 10, we considered those to have limited power (potentially causing weak instrument bias[96]) and removed these from the MR and follow-up analyses.

### Validation of instruments using directionality test
From a drug development point of view, a valid drug will influence the protein level, altering disease risk as a consequence. Therefore, we conducted a directionality test to better understand the direction of effect of the MR findings. We used Steiger filtering[27] to test the directionality of the pQTL-disease associations for all candidate instruments. Any pQTLs with Steiger filter flag as FALSE (which means the pQTL explains more of the variance in the outcome than it does the variance in the exposure) were removed from the MR and follow-up analyses (Tables S1 and S2).

### Validation of instruments using heterogeneity test and specificity test of instruments
Figure S1 illustrates the instrument validation process using a tier system we developed in a previous MR study.[6] We conducted two types of validations in the European samples and split the pQTLs into three tiers. First, we estimated the heterogeneity of pQTL effects across ARIC[21] and INTERVAL.[5] For pQTLs showing heterogeneous effects across the two studies (defined as p value of pairwise Z-score < 0.001; Table S3), we set them as tier 2 instruments (Table S1). With the assumption that a pQTL effect in one study may highlight a true differential putative causal effect of protein on disease, we kept the tier 2 instruments (heterogeneous instruments) and conducted the MR analysis using pQTL from ARIC and INTERVAL separately. Second, a pQTL associated with multiple proteins means we cannot determine which protein(s) influences disease in a MR setting. We therefore estimate the specificity of the pQTLs in European samples from ARIC. Given ARIC only provided the cis-acting pQTLs, we estimated the specificity of the ARIC pQTLs using INTERVAL pQTLs as a reference panel, in which full GWAS summary statistics in cis- and trans-acting regions were provided. For ARIC pQTLs (and their LD proxies with $r^2$ > 0.8) associated with more than five proteins in the INTERVAL data, we set these pQTLs as tier 3 instruments (non-specific instruments) and kept them from the MR analysis with caution of potential non-specificity.

For African samples, given we have limited data to conduct heterogeneity and specificity tests using additional African samples directly, we were not able to conduct the above validation analyses. These tests need to be carefully considered once more African datasets were available).

## Outcome selection in the global biobank meta-analysis initiative
We selected disease GWASs from GBMI using four criteria:

1. Both African and non-Finnish European GWAS summary statistics were available in GBMI.
2. Number of cases over 100 so that the logistic mixed model, SAIGE,[97] used for the GWAS provided good power.
3. the pQTL data we applied were obtained from both males and females, we used sex-combined disease GWAS for the main MR analysis, so the exposure and outcome of the MR were equally represented in the population.
4. Given we conducted sex specific proteome MR analysis as a follow-up analysis, we therefore selected the male- and female-only disease GWAS from GBMI and used them as outcomes in the sex specific MR analysis.

Based on these criteria, eight diseases were selected as the outcomes for the MR analysis, including: idiopathic pulmonary fibrosis (IPF), primary open-angle glaucoma (POAG), heart failure (HF), venous thromboembolism (VTE), stroke, gout, chronic obstructive pulmonary disease (COPD) and asthma. The sample size of the eight African-specific GWASs were from 3,867 to 35,209 (Table S6A). The sample size of the eight European-specific GWASs were from 469,078 to 1,219,215 (Table S6B).

## Discovery proteome-wide MR of complex diseases in European and African ancestries
In the discovery MR analysis, we estimated the putative causal effects of proteins on the eight selected human diseases in European and African ancestries separately. To best represent the genetic signals in the cis-acting region and boost power, we conducted one of the three sets of analysis depending on how many protein instruments had been selected. For proteins with only one instrument, we conducted Wald ratio analysis[29] to estimate the effects between proteins and diseases. For proteins with two or more instruments, we used the conditional independent pQTLs as genetic instruments and applied a generalized inverse variance weighted (gIVW) approach that takes into account the correlation between nearby pQTLs.[30] For proteins with three or more instruments, we further applied a generalized MR-Egger regression (gEgger) approach that considered the correlation among pQTLs.[30] The MR estimates with FDR corrected p value <0.05 were used to select candidate protein-disease signals for follow up analyses

 **CellPress**

**Cell Genomics**

(number of tests for MR in Europeans = 11,612; number of tests for MR in Africans = 9,858). The MR analyses were conducted using the MendelianRandomization R package[89] and TwoSampleMR R package (github.com/MRCIEU/TwoSampleMR).[88]

To select MR estimates with good genetic evidence, we applied two thresholds here:

1. A Benjamini-Hochberg false discovery rate (FDR) of 0.05 was applied to select best MR estimates with robust signals.
2. A MR p value of 0.05 was applied to create an extensive list of MR signals for the sensitivity and validation analysis.

### Sensitivity analysis of candidate MR signals

To increase the reliability of the MR signals, we applied a set of five sensitivity analyses for the candidate MR signals.

### Estimation of horizontal pleiotropy and heterogeneity of MR signals

With increasing power of the protein GWAS, 79.2% of the pQTLs have two or more instruments in the cis region, we therefore applied two sensitivity analyses for conventional Mendelian randomization.

First, we applied the gEgger method[30] and considered the intercept term of the gEgger approach as an indicator to estimate the potential effect of pleiotropy.[31] For MR signals with a gEgger intercept p value lower than 0.05, we considered these protein-disease signals as influenced by horizontal pleiotropy. Due to the importance of controlling for pleiotropy in MR analysis, these MR signals were excluded from any of the follow-up analyses and listed separately in Table S9.

Second, we applied Cochran's Q test for gIVW results and Rücker's Q test for gEgger results[33,34] to estimate the potential heterogeneity of MR estimates across each pQTL.[11] Heterogeneity could be caused by various reasons (e.g. by measurement error),[11] we therefore still kept the MR signals with evidence of heterogeneity for the follow-up analyses, but flag the potential heterogeneity in the MR results tables (Tables S7 and S8).

### Genetic colocalization analysis of the candidate MR signals

Results that passed the MR p-value threshold of p < 0.05 and the pleiotropy test using MR-Egger regression were evaluated using genetic colocalization analysis. The purpose of this analysis was to distinguish putative causal MR signals from protein-disease pairs confounded by LD (see Figure 5A). We applied three sets of colocalization analyses to obtain more reliable colocalization evidence. First, we applied an approximate colocalization analysis we developed, which is noted as LD check.[6] We estimated the LD $r^2$ between each pQTL against all variants with GWAS $P < 1 \times 10^{-3}$ in the region associated with the disease outcomes. $R^2$ of 0.7 between the pQTL and any of the outcome variants was used as evidence for approximate colocalization.

Second, we applied conventional genetic colocalization analysis using the 'coloc' R package.[35] For these colocalization analyses, we used slightly more relaxed prior probabilities that a variant is equally associated with each phenotype ($p_1 = 1 \times 10^{-3}$; $p_2 = 1 \times 10^{-3}$) and both phenotypes jointly ($p_{12} = 1 \times 10^{-4}$). There are two reasons for this: (i) the pQTLs have passed our instrument selection and validation, therefore have good instrument strength to suggest that these variants were robustly associated with the protein level, so we relaxed the prior probability for $p_1$; (ii) as this analysis is based on candidate MR signals, there is some evidence to support the effect of proteins on diseases already, so we relaxed the probability for $p_2$ and $p_{12}$. A colocalization probability (PP.H4) > 70% in this analysis would suggest that the two genetic association signals are likely to colocalize within the test region.

Third, conventional colocalization may provide unreliable inference in some regions due to the presence of multiple independent (but partially correlated) genetic association signals. We therefore applied pairwise conditional and colocalization (PWCoCo) analysis[6] of all conditionally independent pQTLs against all conditionally independent association signals for the disease outcomes. For the 830 and 388 protein-disease pairs showing suggestive MR evidence in European and African ancestries (Tables S7 and S8), we conducted PWCoCo analysis using our newly developed C++ pipeline (https://github.com/jwr-git/pwcoco). The 1000 Genome genotype data for European and African samples were used separately as the LD reference panel[32] for the PWCoCo analysis.

### Estimation of potential aptamer binding artificial effect of pQTLs

The aptamer binding artefacts driven by protein-altering variants may create false genetic associations between genetic variants and proteins and therefore bias the putative causal estimates. We considered the influence of such bias by checking whether the pQTL instruments or their LD proxies ($r^2 > 0.8$) were defined as missense, stop-lost or stop-gained variants using the Ensembl Variant Effect Predictor (VEP)[37] (variants annotation listed in Tables S1 and S2). When the MR signals were identified as involving one or more of these coding variants, we flagged the MR signals to warn the reader of this potential bias (Tables S7 and S8).

### Validation MR analysis

In the validation MR analysis, for any protein-disease pairs that passed the MR threshold p < 0.05, we selected conditionally independent pQTLs from two independent proteome GWAS studies: European pQTLs from Zheng et al[6] and African pQTLs from AASK study. For consistency, the same instrument selection and validation was applied. After selection, there were 285 pQTLs of 285 proteins in Europeans from Zheng et al[6] (Table S4) and 290 pQTLs of 290 proteins in 467 Africans from the AASK study[21] (Table S5). The same MR pipeline used for the discovery MR was applied here for the validation MR. A Benjamini-Hochberg FDR of 0.05 was applied to pick out MR estimates with validation MR evidence.

### Sex-specific MR of candidate protein-disease signals

We conducted sex-specific MR analysis to identify protein-disease pairs with different effect estimates in males and females. All proteins and disease outcomes were included in this analysis. Among them, 8649 protein-disease pairs and 8527 pairs have available data in European ancestry to conduct the MR analysis in males and females separately, similarly 8076 and 7851 pairs have available data for African ancestry. The sex-specific GWAS of the eight outcomes were used as outcomes for the sex-specific MR. The same statistical model and sensitivity analyses pipeline was applied in the sex specific MR. A pair-wise Z score p value less than 0.01 between male- and female-only MR estimates was used as threshold to pick out MR estimates with sex specific effects (Table S11).

### Multi-ethnic comparison of pQTL and proteome MR effect estimates across ancestries
#### Estimation of ancestry specificity of pQTL and MR effects across ancestries

We systematically evaluated the ancestry specificity of pQTL using two functions implemented in the TAMR package (https://github.com/universe77/TAMR): (i) estimate whether the direction of effect was consistent across ancestries; (ii) estimate whether the signals was significant in both ancestries. To better control the influence of different effects, allele frequencies and power of pQTLs across the two ancestries, we applied a Bayesian Winner's Curse correction analysis described in a previous study.[39] The method estimates probability that one pQTL has a matching pQTL across two ancestries using beta, se and sample size of the pQTLs in the two ancestries and further estimates the expected number of pQTLs with same direction of effect and/or same level of significance. For the 1,096 proteins with full summary statistics available, we first excluded proteins showed no pQTL association in either ancestry. For the remaining 1,076 proteins, we conducted the ancestry specificity analysis using the European pQTL effects to mimic African pQTL effects and vice versa. In total, four analyses were conducted:

1. African pQTLs estimate the direction of effect of European pQTLs (Table S12A).
2. African pQTLs estimate the significance of African pQTLs (Table S12A).
3. European pQTLs estimate the direction of effect of African pQTLs (Table S12B).
4. European pQTLs estimate the significance of African pQTLs (Table S12B).

#### Estimation of multi-ancestry and ancestry-specific pQTLs

After obtaining an overall idea of the ancestry specificity of pQTLs, we generated a list of multi-ancestry and ancestry specific pQTLs using the following approach:

1. For regions with pQTL signal in one ancestry but not the other, we defined these regions as ancestry specific regions (Figure 3 situation 4) and set pQTLs in these regions as ancestry specific pQTLs in non-shared regions (Table S13).
2. For the remaining regions with pQTL signals in both ancestries, we looked up the European pQTLs (or its LD proxies with LD $r^2 > 0.8$) in the African ancestry and vice versa. If there is overlap signal (with $P < 1 \times 10^{-3}$), we set them as multi-ancestry pQTLs (Figure 3 situation 2 or 3; Table S14).
3. For the regions with pQTLs in both ancestries, we further picked out those pQTLs without validation signal in the other ancestry and noted them as ancestry specific pQTLs in the shared regions (Figure 3 situation 1; Table S15).

#### Estimation of multi-ancestry and ancestry-specific proteome MR signals

We conducted a multi-ancestry comparison of the proteome MR estimates across European and African ancestries. To identify the multi-ancestry protein-disease pairs with MR evidence in both ancestries, we selected the 60 pairs that passed FDR threshold of 0.05 in the discovery MR analysis (Tables S7A and S8A). Within the 60 pairs, 59 of them were unique pairs, with the ABO level on VTE showed strong MR signals in both ancestries. For the 59 unique pairs, we corrected their MR signals using an FDR threshold of 0.05 based on the 59 pairs. For those pairs that passed the FDR threshold in the multi-ancestry comparison analysis (Table S16A), we further checked their MR and colocalization evidence in the validation MR analysis. Those pairs with MR and colocalization evidence in both multi-ancestry comparison and validation analysis was considered as the most reliable multi-ancestry protein-disease signals.

We also tried to identify ancestry specific protein-disease pairs that only showed MR evidence in one ancestry (but not in the other). For African specific MR signals, we selected 86 protein-disease pairs that showed marginal MR signal (MR p < 0.05) and colocalization evidence (colocalization probability>0.7) in the discovery MR analysis. We applied an FDR threshold of 0.05 based on these 86 pairs. For those pairs passed the FDR threshold, we checked their effect in the European proteome MR and excluded pairs that showed up in the multi-ancestry MR list (Table S16A). For the remaining pairs, we checked their MR and colocalization evidence in the African validation MR (Table S16B). Those protein-disease pairs showed MR and colocalization evidence in both African specific analysis and validation analysis were picked as the African specific protein-disease signals.

For European-specific MR signals, we applied the same approach as the African specific analysis. This analysis was conducted by selecting 386 protein-disease pairs with marginal MR evidence (MR p < 0.05) and colocalization evidence (colocalization probability>0.7) in the discovery MR analysis. The same FDR correction was applied, with protein-disease pairs overlapped with the multi-

ancestry list been excluded (Table S16C). The MR and colocalization evidence from the European validation analysis were further considered. Those pairs with MR and colocalization evidence in European specific analysis and validation analysis were selected as the European specific protein-disease signals.

### Triangulation of protein-disease MR signals with observational and clinical trial evidence

Proteins are the targets for most drugs, and therefore have high value for drug target validation and drug reposing. For the 16 protein-disease pairs with multi-ancestry or ancestry-specific evidence in the multi-ancestry comparison analysis, we triangulated these MR findings with the observational evidence obtained from HUNT (Table S17) as well as with clinical trial evidence provided by the Open Targets[42] and DrugBank[43] databases (Table S18).

For these 16 protein-disease pairs, we estimated their observational associations using individual level data from HUNT. The protein measurements for the 16 disease-associated proteins were rank transformed using the function RankNorm in the RNOmni R package. Further, residuals were extracted from a linear model including the transformed protein values in addition to age and sex. The residuals were then included in a logistic model for disease status in addition to age and sex as covariates. The eight disease outcomes (Asthma, COPD, Gout, POAG, VTE, IPF, Stroke, HF) were defined according to the GBMI phecodes using ICDs (details in the GBMI flagship paper[23]). All individuals provided informed written consent and the study was approved by the Regional Committee for Medical and Health Research Ethics (REK # 2018/1622).

In addition, since the HUNT proteome datasets were collected in a cardiovascular disease (CVD) enriched cohort, we considered the potential influence of sample selection on the observational associations and used the prevalent cases vs controls as the model (Table S17).

For Open Targets, the individual score of each evidence category was recorded (Table 2). For proteins targeting existing drugs or drugs under clinical development, we further checked the details of the clinical trials from Clinialtrials.gov and recorded drug names and primary indications (e.g. a disease). In DrugBank, each protein was searched as a target. The protein-drug pair with actual action (e.g. as an inhibitor) was recorded together with the primary indications. For observational associations, the direction and robustness of the association signals of MR and observational analyses were compared (Table 2 and Table S18).

### Drug target prioritization profiling

To summarise evidence of this study, we developed a drug target prioritization profiling procedure. Four key steps were used here to select the most promising protein-disease pairs from 11,612 pairs in European ancestry and 9,858 pairs in African ancestry (Figure 6A). First, the MR signals with FDR <0.05 in discovery (or multi-ancestry comparison analysis) were used to select candidate protein-disease pairs. Second, sensitivity analyses including pleiotropy test (little evidence from MR-Egger regression) and three types of colocalization (colocalization probability>0.7) were applied to select more robust protein-disease pairs. Third, we further selected protein-disease pairs with multi-ancestry or ancestry-specific MR evidence (FDR<0.05 in multi-ancestry comparison) and validation MR evidence, and considered them as protein-disease pairs with reliable genetic evidence. Fourth, integrating the genetic evidence with observational and trial evidence, the most reliable pairs with high drug development value were prioritized. The evidence level for the most promising pairs were summarized in Figure 6B and Table S18.

