## [Document S2. Transparent peer review records for Zhao et al. · Cell Genomics]

Proteome-wide Mendelian randomization in global biobank meta-analysis reveals multi-ancestry drug targets for common diseases

Huiling Zhao¹, Humaria Rasheed^{1,2,3}, Therese Haugdahl Nøst^{2,4}, Yoonsu Cho¹, Yi Liu¹, Laxmi Bhatta², Arjun Bhattacharya^{5,6} Global Biobank Meta-analysis Initiative, Gibran Hemani¹, George Davey Smith^{1,7}, Ben Michael Brumpton^{3,8,9,16}, Wei Zhou^{10,11,12,16*}, Benjamin M. Neale^{11,12,16*}, Tom R. Gaunt^{1,7,16*}, Jie Zheng^{1,13,14,15,16*}

Summary

Initial submission: Received : 1/15/22

Scientific editor: Emily Marcinkevicius and Laura Zahn

First round of review: Number of reviewers: 3
Revision invited : 3/14/2022
Revision received : 6/16/2022

Second round of review: Number of reviewers: 3
Accepted : 9/21/2022

Data freely available: Yes

Code freely available: Yes

This transparent peer review record is not systematically proofread, type-set, or edited. Special characters, formatting, and equations may fail to render properly. Standard procedural text within the editor's letters has been deleted for the sake of brevity, but all official correspondence specific to the manuscript has been preserved.

Referees' reports, first round of review

Reviewer#1

The authors assessed causal effects of proteins on eight traits and diseases by conducting two-samples MR using publically available summary statistics, followed by colocalization, and linking the results to data from clinical trials and observational studies. One of the many highlights is the inclusion of both European and African ancestry datasets. The manuscript is very well written, methodologically sound, and the results are relevant and new. In addition, the work provides a blueprint for conducting multi-ancestry MR and colocalization analyses. It was a real pleasure to read this manuscript. Thus, I have only a few minor comments listed below for consideration.

1. It would be interesting to know how many (or percentage) of pQTLs were removed by Steiger filtering (could be added e.g. in Figure 1). I could not find this information in Tables S1 and S2.
2. As far as I understood, no sex-specific pQTLs were available and included in the sex-specific MR. Does this influence the validity of the sex-specific MR? I suggest to comment on this briefly in the Discussion.
3. A short note what "conditionally independent" pQTL means (i.e. what the condition was) would be helpful for the reader.
4. Results p. 5 "Discovery MR": According to Table S7a, shouldn't it read 59 instead of 69 EA MR FDR<0.05 results?
5. Results p. 9, first paragraph (similar issue): According to Table S16C, there are seven EA replicated MR results (not eight).

Reviewer#2

In this article, the authors systematically estimated the causal role of 1,311 and 1,310 proteins, measured in populations from African and European ancestry respectively (1,076 proteins in both ancestries), on eight complex diseases using a comprehensive ancestry-specific MR pipeline based on our previous approach in European datasets⁵. They further estimated the consistency of pQTLs across ancestries, identified potential multi-ancestry and ancestry-specific causal protein-disease pairs, and integrated MR findings with observational and clinical trial evidence²³ to prioritize drug targets. The study is comprehensive. I have following concerns.

1. This study primarily used MR methods to scan for protein-disease pairs. In preprocessing steps, the authors first conducted LD clumping to remove SNPs with large correlations and then used F-statistics to select the strong instrument variants (IVs). Later, they applied gIVW and gEgger to account for LD. First, the field of MR advances very fast. Both gIVW and gEgger are methods that cannot account for weak IVs and LD. gIVW cannot account for horizontal pleiotropy while gEgger makes strong assumptions on equal effects of pleiotropy. Many MR methods have been developed for those issues. Recently, several methods have been proposed to account for correlated horizontal pleiotropy. But the authors chose two "old" methods that cannot account for those issues. It introduces a problem of using so less IVs in the analysis. For majority of protein-disease pairs, the number of IVs used was quite small, largely reduced the replicate rates. It will be helpful that the authors provide some rationale and discussions to choose gIVW and gEgger among them.
2. The first assumption for MR analysis is that IVs associated with the exposure, but not necessarily to be causally. Regarding paragraph on page 6, the step to annotate the IVs as missense and others is not necessary as we do not need IVs to be causal variants for any protein levels.
3. Some of results in the main text are not clear. For example, in the subsection of "Identification of multi-ancestry and ancestry specific pQTLs", most of the paragraph describes Fig. 3 and

related tables. I cannot tell what messages the authors want to deliver. The same thing happens for the second paragraph on page 9. The whole paragraph looks like a caption for Fig. 4.

4. Intuitively, the protein expression level could be different in case group and control group, using the summary statistic largely from control group may bias the causal estimates. The authors should provide rationale and discussions over this choice.

5. In the data availability section, I cannot find download links for summary data used.

Reviewer#3

I want to congratulate the authors for this very thorough analysis that aim to identify "putative" causal genes on a series of medical conditions with high unmet clinical need. The group of authors have extensive track record in this area. This group once more takes the field of MR one step forward. Below, some comments with the goal of improving the clarity and quality of this exceptional work.

Introduction:

1. First paragraph of this paper is described towards "pharmacogenetics", which though also involves drugs and genes is slightly different from the authors are attempting to do. To use human genetics to identify and validate drug targets for a wide range of disorders. Indeed the Ref #1 is about Pharmacogenetics (PGX), please modify the paragraph 1 and ref-1 accordingly. There are obvious reasons for using multi-ancestry for drug-discovery. e.g. LOF mutation in PCSK9 from African-American (NEJM) lead the way for drug-target discovery.

2. Conceptual clarification: given the resources they are using (outcomes being prevalent plus incident(could not see this but is usual GWAS?)): This type of analysis is only relevant to identify drug-targets for primary prevention of the conditions described in this manuscript. Drug target identification for 2ry or acute management requires a different setting, which I assume some of the co-authors are well aware.

3. With a few exceptions authors talk of "causal" evidence through the entire manuscript. I suggest they temper their language and use "putative" causal. This will match more closely with the large battery of analyses they conducted, which is clear evidence of the various assumptions and limitations of the strong strategy they applied. Indeed some of the assumptions of IV are unverifiable, something the authors are well versed.

Results / methods:

4. Clarification of the decision making process: Through the manuscript the authors used word to denote some categories/degree of credibility in their findings. Some examples I identified are: (a) Fig 6. they talk of "robust/ moderate". (b) Little evidence of horizontal pleiotropy. Please clarify this statements to secure reproducibility of their findings/strategy.

5. pQTL sources/replication: I wonder if possible to describe the overlap among the different sources of pQTLs. Plus, as I Somascan user, I am aware those sources are from different versions, which add some degree of technical difficulty for interoperability. Can the authors comment on this? specially as the pQTLs used for discovery and replication comes from wide variety of SOMAscan version. A question of clarification: is the use of alternative sources of pQTL a replication or a sensitivity analysis? To this reviewer the sensitivity analysis seems more appropriate (at least I am missing something).

6. Quality of the Phenotypes: there is no mention as a weakness that the phenotypes used are just Phecodes. These are just ICD-only phenotypes without "clinical chart review" adequate for this type of large-scale discovery process, but clearly not ideal if the implications are to initiate/continue discovery programs and trials with large-financial implications, please acknowledge this as a limitation.

7. Findings with $p < 0.05$ but $FDR > 5\%$: please remove these findings. This goes against the very strict steps/approach authors used to minimize any false positive findings.

8. Triangulation of evidence: I like the terminology and endorse the use of complementary sources of evidence. However, I have some strong reservations to the ones used by the authors. Observational: some of the authors of this paper have wrote extensively about the limitations of observational evidence as the reason for MR to exist. Now I am surprise how now they re-

introduce this evidence, and even categorize as "moderate" only after logistic regression with age and sex as co-variates. This reviewer disagree with this statement and propose to exclude this. Also please clarify what exactly is the "extra" evidence provided by trial.

9. Novelty: can the authors clarify the process to declare novelty? Do they use other MR - GWAS-pQTLs recently published or GWAS-catalogue.

10. LD-check: I really like this, statistical colocalization does not remove confounding by LD. However, can the authors clarify what is exactly the trait/s they checked against? I was under the assumption they were checking for the same or related trait as the one the MR discovered, please clarify. Also please clarify how did you solve the problem of GWAS using different terminology for phenotypes.

11. Recommendations: I will assume that other groups will have slightly different variations of the recommendations. I have a few to start with: (a) The authors of MR should clearly specify what is the target trial they are attempting to emulate. (b) efficacy outcomes should be those used in the trial they are attempting to emulate or as close as possible. (c) MR can also inform on on-target safety events (reason as to which 30% of the new drug-discovery programs failed). I will call (a) and (b) in-silico trials. (d) horizontal pleiotropy: for cis-MR where is very rare to have >3 instruments or ideally 10 to run MR Egger, HP is the main problem. Despite that this is a topic whose conditions are clearly defined, the approaches most people take (including this group) are approximations (e.g. instrument >5 proteins, or test for heterogeneity) are not indeed test for horizontal pleiotropy.

Authors' response to the first round of review

Reviewer #1: The authors assessed causal effects of proteins on eight traits and diseases by conducting two-samples MR using publicly available summary statistics, followed by colocalization, and linking the results to data from clinical trials and observational studies. One of the many highlights is the inclusion of both European and African ancestry datasets. The manuscript is very well written, methodologically sound, and the results are relevant and new. In addition, the work provides a blueprint for conducting multi-ancestry MR and colocalization analyses.

It was a real pleasure to read this manuscript. Thus, I have only a few minor comments listed below for consideration.

Thanks for the positive feedback from Reviewer 1. We appreciated the nice suggestions from you. We have added additional information requested by the reviewer in this revision.

1. It would be interesting to know how many (or percentage) of pQTLs were removed by Steiger filtering (could be added e.g. in Figure 1). I could not find this information in Tables S1 and S2.

Reply: Thank you for pointing out this issue. We have calculated the number of pQTLs in each step and summarized it in the following tables. For European ancestry, the percentages of pQTLs that failed to pass Steiger filtering (indicating wrong direction of effects from outcomes to exposures) for the eight disease outcomes range from 0.5% ([4894-4870]/4894=0.5%) to 6.5% ([4896-4577]/4896=6.5%). For African ancestry, the percentage of pQTLs that failed to pass Steiger filtering for the eight outcomes are from 0.7% ([3173-3151]/3173=0.7%) to 23.4% ([3162-2421]/3162=23.4%). We have added Steiger filtering as a selection process to Figure 1. We have added these results into two additional tables, Table S1B and S2B.

Also, we filtered pQTLs based on their instrument strength (after LD clumping). Around 11.5% ([5974-5285]/5974=11.5%) of SNPs have been removed due to potential weak instrument bias

(indicated as *F*-statistics < 10).

Table S1B

European ancestry conditional snp: 7650(7105 pQTLs)(1992 protein)							
after mapping with EA summary data: 6614(6107 pQTLs)(1311 protein)							
after 0.6 LD clumping: 6144(5974 pQTLs)(1310 protein)							
Removing weak instruments (F-statistics<10) 5418(5285 pQTLs)(1310 protein)							
match with Asthma outcome	match with COPD outcome	match with Gout outcome	match with POAG outcome	match with VTE outcome	match with IPF outcome	match with Stroke outcome	match with HF outcome
4894 pQTLs (1297 protein)	4895 pQTLs (1297 protein)	4896 pQTLs (1297 protein)	4893 pQTLs (1297 protein)	4894 pQTLs (1297 protein)	4896 pQTLs (1297 protein)	4895 pQTLs (1297 protein)	4898 pQTLs (1297 protein)
after steiger filtering	after steiger filtering	after steiger filtering	after steiger filtering	after steiger filtering	after steiger filtering	after steiger filtering	after steiger filtering
4870 pQTLs (1294 protein)	4774 pQTLs (1288 protein)	4774 pQTLs (1288 protein)	4729 pQTLs (1290 protein)	4744 pQTLs (1291 protein)	4577 pQTLs (1278 protein)	4749 pQTLs (1289 protein)	4824 pQTLs (1294 protein)

Table S2B

African ancestry conditional snp: 4246(4129 pQTLs)(1605 protein)							
after mapping with AA summary data: 3900(3793 pQTLs)(1312 protein)							
after 0.6 LD clumping: 3875(3785 pQTLs)(1311 protein)							
Removing weak instruments (F-statistics<10) 3550(3473 pQTLs)(1311 protein)							
match with Asthma outcome	match with COPD outcome	match with Gout outcome	match with POAG outcome	match with VTE outcome	match with IPF outcome	match with Stroke outcome	match with HF outcome
3173 pQTLs (1263 protein)	3173 pQTLs (1263 protein)	3173 pQTLs (1263 protein)	3050 pQTLs (1246 protein)	3173 pQTLs (1263 protein)	3162 pQTLs (1261 protein)	3168 pQTLs (1263 protein)	(1263 protein)
after steiger filtering	after steiger filtering	after steiger filtering	after steiger filtering	after steiger filtering	after steiger filtering	after steiger filtering	after steiger filtering
3151 pQTLs (1263 protein)	3108 pQTLs (1254 protein)	3070 pQTLs (1252 protein)	2790 pQTLs (1204 protein)	3083 pQTLs (1249 protein)	2421 pQTLs (1132 protein)	3041 pQTLs (1253 protein)	3089 pQTLs (1251)

2. As far as I understood, no sex-specific pQTLs were available and included in the sexspecific MR. Does this influence the validity of the sex-specific MR? I suggest to comment on this briefly in the Discussion.

Reply: Thank you for the nice suggestion. The reviewer is right that sex-specific pQTLs are not publicly available yet. Therefore, we used the sex-combined pQTL data plus the sex-specific outcome data for the sex-stratified analysis in this study. This is under the assumption that protein levels were similar across males and females. We expect this assumption will be valid in most cases, as the recent Fenland pQTL paper stated that of 412 protein-disease pairs, only 10 (2.43%) showed sex-specific pQTL effects (Pietzner et al., 2021)

To reflect this, we have added the following paragraph in the Discussion section:

“However, due to a lack of available sex-specific pQTLs, our sex-specific MR was conducted using sex-specific outcome data only. This is under the assumption that genetic effects of pQTLs are generally similar between males and females. A recent proteome GWAS suggested that only 2.43% proteins showed sex-specific pQTL effects⁷, which supports this assumption. However, whether the prioritised targets and/or drugs may have different drug responses in males and females needs further investigation.”

3. A short note what "conditionally independent" pQTL means (i.e. what the condition was) would be helpful for the reader.

Reply: Thank you for the suggestion. We have carefully checked the conditional analysis section conducted by Zhang et al and added the following explanation in the section “Summary of selection and validation of proteins and diseases data”:

“We defined conditionally independent pQTLs as a set of cis-acting pQTLs that have independent genetic effects on the tested protein. The conditionally independent pQTLs were identified using the forward backward stepwise regression implemented in the QTLtools package (<https://qtltools.github.io/qtltools/>), followed by the fine mapping method SuSiE²⁵.”

4. Results p. 5 "Discovery MR": According to Table S7a, shouldn't it read 59 instead of 69 EA MR FDR<0.05 results?

Reply: Thank you for this detailed suggestion. In European ancestry, we got 69 MR results with FDR<0.05. but 10 of them didn't pass the pleiotropy test. Since absence of pleiotropy is a key assumption for MR analysis, we only present 59 results in Table S7A. To be clearer, we have revised the Table S7A title as "Protein-disease associations that passed FDR < 0.05 and pleiotropy test in European ancestry". We have also revised the main text as follow:

"We conducted discovery MR on 10,318 protein-disease pairs in European ancestry and 9,858 pairs in African ancestry. Among these, 69 MR signals in European ancestry and 1 signal in African ancestry reached a Benjamini-Hochberg false discovery rate (FDR) of 0.05.

.....

As pleiotropy invalidates the exclusion restriction assumption of MR, these 10 results were excluded from the candidate putative causal effects list. 59 MR signals in European ancestry and one signal in African ancestry were therefore considered as candidate protein-disease pairs in the discovery analysis (Table S7A and S8A)."

5. Results p. 9, first paragraph (similar issue): According to Table S16C, there are seven EA replicated MR results (not eight).

Reply: Thanks for this kind remainder. We have revised the number to seven in this revision. Reviewer #2: In this article, the authors systematically estimated the causal role of 1,311 and 1,310 proteins, measured in populations from African and European ancestry respectively (1,076 proteins in both ancestries), on eight complex diseases using a comprehensive ancestry-specific

MR pipeline based on our previous approach in European datasets⁵. They further estimated the consistency of pQTLs across ancestries, identified potential multi-ancestry and ancestry-specific causal protein-disease pairs, and integrated MR findings with observational and clinical trial evidence to prioritize drug targets. The study is comprehensive. I have following concerns.

1. This study primarily used MR methods to scan for protein-disease pairs. In preprocessing steps, the authors first conducted LD clumping to remove SNPs with large correlations and then used F-statistics to select the strong instrument variants (IVs). Later, they applied gIVW and gEgger to account for LD. First, the field of MR advances very fast. Both gIVW and gEgger are methods that cannot account for weak IVs and LD. gIVW cannot account for horizontal pleiotropy while gEgger makes strong assumptions on equal effects of pleiotropy. Many MR methods have been developed for those issues. Recently, several methods have been proposed to account for correlated horizontal pleiotropy. But the authors chose two "old" methods that cannot account for those issues. It introduces a problem of using so less IVs in the analysis. For majority of protein-disease pairs, the number of IVs used was quite small, largely reduced the replicate rates. It will be helpful that the authors provide some rationale and discussions to choose gIVW and gEgger among them.

Reply: We thank you reviewer 2 raise this important comment. We have considered this from three angles:

1. For this study, we focused on protein QTLs in the cis-acting region, since these variants are more likely to have a direct biological link with the tested proteins. When considering MR methods to apply, we carefully reviewed most of the recent cis-MR methods, such as LD clumping, conditional analysis (e.g. CoJo), principal components (PCA) analysis, factor analysis and Bayesian variable selection (Gkatzionis et al., 2021). Our instrument selection approach integrated three of the above approaches: LD clumped IVs to $r^2 < 0.8$; conditional/fine mapped genetic variants (results created using SuSiE from the original

GWAS paper); as well as a PCA approach as discovery (gIVW/gEgger). We believe such integration provides reliable IVs and powerful MR estimates. The Bayesian methods were not applied here since they can only provide probability rather than an estimate of causal effect size.

2. Re MR methods that account for correlated pleiotropy, we thank the reviewer for raising this important point. Some methods such as LCV, CAUSE, cML-MA, LHC-MR and MRAPSS have been developed in this area. Our recent review of MR methods suggested the CAUSE method (Morrison et al., 2020) as an outlier/variant adjustment that allows some directional pleiotropy (Sanderson et al., 2022). In more detail, these methods used genomewide variants to estimate causality, which is an important innovation from a MR methodological point of view. Applying these methods to omics data will be a very exciting extension of these methods. However, by carefully checking the citations of CAUSE, we found that the method was mostly applied to classical risk factors represented by many GWAS signals across the whole genome (e.g. blood cell traits/BMI etc.). Whether these methods can be applied to a specific genomic region (e.g. the cis-acting region) needs to be properly evaluated before large-scale application in a study such as ours. As a first attempt at this, we validated our eight prioritised pQTL MR findings (Table S18) using all cis-acting variants provided by Zhang et al. and tested against the putative causal outcomes using the CAUSE method. Among the eight protein-disease pairs, IL7R has no instrument in African ancestry, and was only tested in European ancestry. The remaining seven pairs were tested in both European and African ancestries (14+1=15 tests). For these 15 tests, gIVW showed MR evidence for 11 of them at $P < 0.05$. In comparison, CAUSE only suggested four of the 11 (36.4%) tests as significant, which implies that CAUSE has lower power than gIVW (more discussion in point “b” below). When comparing the direction of effect, 12 of them showed the same direction among the two methods. The remaining three tests showed little evidence of causal effect using both methods. In addition, CAUSE suggested that eight of the 15 tests are more likely to be causal than sharing (delta_elpd is less than zero, which indicate that the posteriors estimated under the causal model fit the data better than posteriors estimated under the sharing model). All these eight tests showed MR evidence at $P < 0.05$ using gIVW. The four tests with little gIVW evidence have little evidence to support their causality than sharing. Collectively, 12 of the 15 (80%) CAUSE causality estimates fit with the gIVW results. This provides additional evidence to support the reliability of our findings. During the application of CAUSE, we observed two situations, which may provide useful recommendations for the CAUSE method in omics MR.

a. SNP coverage: the CAUSE method is designed to use genetic variants across the whole genome. The default setting requires one million (or more) genetic variants to be sampled to estimate the nuisance parameters, which is used to define the prior distribution of the model. The tips in the documentation suggested not to estimate the nuisance parameters with substantially fewer than 100,000 variants, which can lead to poor estimates of the mixing parameters and bad model comparisons. However, for the pQTL data we used, we only have access to an average of 6,000 variants per protein in the cis-acting regions, which is much less than the required number of variants or genomic coverage. Therefore, we got the warning message “In est_cause_params(X, varlist) : Fewer than 100,000 variants are being used to estimate parametrs. This can cause problems and is not recommended”, which highlighted the SNP coverage issue. If we consider using other proteome datasets with whole genome coverage, e.g. DeCODE proteome data, we still need to

carefully consider the following questions:

i. DeCODE (and most of the other available datasets) only have data in European ancestry, which did not fit with our study design of multi-ancestry comparison.

ii. the difference between cis- and trans-acting pQTLs is complicated and needs to be tested case by case. For example, APOB showed consistent effects across cis- and trans-acting variants, which could be a good fit for the CAUSE method. For other proteins, such as IL6R, the GWAS signals mainly exist in the cis region, which may not fit the CAUSE assumption.

b. Power/scope of CAUSE: only 36.4% of the prioritised protein-disease pairs showed marginal significance using the CAUSE method. All of them showed weaker associations than gIVW. We considered two possible reasons:

i. SNP coverage, which has been discussed above

ii. the power of the CAUSE method is moderate. In the CAUSE paper, the authors stated that CAUSE has somewhat lower power than other MR methods (e.g. IVW) in most settings, especially when the outcome data has low power. Therefore, we consider CAUSE as a useful sensitivity method to reduce false positive rate (when we have good SNP coverage and good power of GWAS data), whilst IVW is a good option as a discovery method.

In summary, CAUSE is a useful method to consider correlated and uncorrelated horizontal pleiotropy using whole-genome variants as instruments. However, due to the current limitations of the method, it might not be suitable to systematically apply CAUSE as a discovery method for omics MR. To summarize the above observations, we have added a paragraph in the discussion / recommendation section to propose future methods development of CAUSE for omics MR.

“Some recent methods such as CAUSE account for correlated and uncorrelated pleiotropy using genome-wide summary statistics⁸⁰, but are not optimized for analysis in a specific genomic regions. With more methods development (e.g. a CAUSE based method in the cis-acting region) and evaluation of other methods such as contamination mixture methods⁸¹ and MR locus⁸², such approaches may offer the potential to estimate level of pleiotropy in the near future.”

3. Re the comments of number of IV, this is a common feature of omics traits such as protein levels, where cis-acting QTLs have a larger and more direct effect, whilst trans effects are typically polygenic with many small effects. Based on current sample sizes, the majority of genetic signals will therefore normally be identified in the cis-acting region. Given this focus on cis-acting regions (rather than signals in the whole genome), the number of conditionally independent signals will normally be limited. To mitigate this, we have included instruments with F -statistics > 10 , which is a relatively inclusive selection criterion to increase the number of IVs in the MR analysis.

2. The first assumption for MR analysis is that IVs associated with the exposure, but not necessarily to be causally. Regarding paragraph on page 6, the step to annotate the IVs as missense and others is not necessary as we do not need IVs to be causal variants for any protein levels.

Reply: We totally agree with reviewer 2 that IVs for MR does not need to be causal, instead they just need to be LD proxies of the causal variants. The reason we conducted a variant annotation for the IVs was because we have concerns there may be binding artefacts caused by variants (a potential issue of SOMA logic technology). Such variant annotation has been widely applied in recent GWAS/MR studies of plasma proteome (Sun et al., 2018a). In fact, we only annotated the

pQTLs to give the reader a warning of these potential artefacts. These missense variants were still included as IVs for the MR analyses, which is in line with reviewer 2's suggestion.

3. Some of results in the main text are not clear. For example, in the subsection of "Identification of multi-ancestry and ancestry specific pQTLs", most of the paragraph describes Fig. 3 and related tables. I cannot tell what messages the authors want to deliver. The same thing happens for the second paragraph on page 9. The whole paragraph looks like a caption for Fig. 4.

Reply: Thank you for providing this feedback to help us improve the presentation of the manuscript. For the section "Identification of multi-ancestry and ancestry specific pQTLs" and Figure 3, we aimed to identify cis-acting pQTLs that showed ancestry specific or trans-ancestry effects on related proteins. The motivation for conducting this analysis is that we would like to understand the reason for identifying ancestry-specific MR effects. Given a protein-disease pair with ancestry-specific

MR evidence (e.g. the effect of CD248 on gout that only showed up in African ancestry) one possibility is that the pQTLs of the protein were different (not shared) across ancestries (e.g. pQTLs of CD248 in Africans and Europeans were not in LD with each other). Alternatively, the pQTLs were shared across ancestries but the outcome signals were very different across ancestries. For example, SERPINF1 showed a strong effect on stroke in African ancestry ($b=-0.54$, $P=3.76 \times 10^{-5}$) but almost no effect in European ancestry ($b=-0.005$, $P=0.83$). By checking the pQTL overlap across the two ancestries, we found that the leading African signal (rs4362) was almost in perfect LD ($r^2=0.96$) with the leading European signal (rs4353; Europeans: $\text{Beta}=-0.746$, $95\% \text{CI}=-0.773$ to -0.719 , $P=1.0 \times 10^{-200}$, Africans: $\text{Beta}=-0.548$, $95\% \text{CI}=-0.608$ to -0.488 , $P=8.8 \times 10^{-66}$; Table S14). This means the leading pQTL showed robust and similar effects on protein level of SERPINF1 in African and European ancestries. Therefore, the difference in stroke risk was likely to be caused by an interaction with an effect modifier for the protein or by different protein levels of SERPINF1 across ancestries. Based on the above motivation, we conducted a systematic analysis to stratify pQTLs into four possible situations. In situation (1), there are pQTL signals in the cis region across the two ancestries, but these pQTLs were not shared (the CD248 case we mentioned above fits this situation). In situation (2) and (3), there is a shared pQTL signal across ancestries (the SERPINF1 example fits this situation), but situation (3) also has a non-shared signal in the same region. In situation (4), there is a pQTL signal in one ancestry but not in the other one. For example, in one of our top MR findings, IL7R effect on asthma, we find IL7R only has pQTLs in European ancestry. To explain these four situations more clearly, we created Figure 3. Figure 3A presents illustrative regional plots to explain the four situations. Figure 3B and C showed the number of proteins/genomic regions been classified in each situation.

To clarify the motivation and the analysis pipeline of the pQTL overlap analysis, we have refined this section in the main text:

"Ancestry specificity of pQTL instruments can be a direct cause of differences in estimated ancestry-specific causal effects of proteins on diseases. Therefore, there is a need to distinguish trans-ancestry and ancestry-specific pQTLs amongst our pQTL instruments. We stratified the 1,310 and 1,311 tested proteins into four possible situations based on the pQTL signals across ancestries (Figure 3A): (1) a protein within a genomic region that has pQTL signals in both ancestries but for which the signals were not shared across ancestries; (2) a protein within a genomic region that has one or more shared pQTLs across ancestries (and without non-shared pQTLs); (3) a protein within a genomic region that has both shared and non-shared pQTLs; (4) a protein within a region that only has pQTL in one of the ancestries. As shown in Figure 3B and 3C, among the tested proteins 1,076 are in regions with pQTLs in both ancestries (situation 1, 2 or 3). The remainder comprise 234 proteins within regions with European-specific pQTLs and 235 proteins within regions with

African-specific pQTLs (situation 4; Table S14A and 14B). For the 1,076 proteins within regions with multiple pQTL signals, 974 had shared pQTLs in the relevant regions (situation 2 or 3; Figure 3C; Table S13). The remaining 102 proteins only had non-shared pQTLs in the regions (situation 1; Figure 3C; Table S15A and 15B)”

We have also refined details of Figure 3 to better present the pQTL overlap analysis.

For the paragraph related to Figure 4, we hope to achieve two aims. First, to summarise the key findings of the multi-ancestry comparison analysis. Second, to select a set of protein-disease pairs as candidates for the drug target prioritisation analysis. We agree with reviewer 2 that part of this section is like a caption of Figure 4. In this revision, we have moved the results summary part to the caption of Figure 4. The paragraph, after this refinement, has been changed in the main text: “As shown in Figure 4, the two multi-ancestry, seven European-specific and seven African-specific signals with MR and colocalization evidence in both multi-ancestry comparison and replication were considered as candidates to be included in the triangulation analysis (Figure 4D).”

Caption of Figure 4:

“Figure 4. Comparison of multi-ancestry proteome MR signals in European and African ancestries. (A) Protein-disease pairs with $FDR < 0.05$ in multi-ancestry comparison. Seven trans-ancestry, 12 African-specific and 89 European-specific candidate protein-disease pairs were selected. (B) Comparison of MR effect estimates of the seven protein-disease pairs with MR evidence ($FDR < 0.05$ in multi-ancestry analysis), each point refers to one protein-disease pair, the x-axis refers to the MR estimate in European ancestry, the y-axis is the MR estimate in African ancestry. (C) Miami plot of the protein-disease causal estimates in European and African ancestries, each point refers to a protein-disease pair, the x-axis is the chromosome and position of the protein, the y-axis is the $-\log_{10}(P)$ of the MR estimate in European (upper) and African ancestry (bottom); the points with colours refer to the seven, 12 and 89 protein-disease pairs with multi-ancestry, African-specific or European-specific MR effects ($FDR < 0.05$ in multi-ancestry analysis), different colour

refers to different outcomes of the protein-disease pairs; the points with legends are the two, seven and seven protein-disease pairs that showed MR and colocalization evidence in discovery and replication MR analyses; background colours in the legends refer to multi-ancestry (yellow), European-specific (green) and African-specific (blue) MR estimates. (D) summary of proteindisease pairs with MR (FDR<0.05) and colocalization evidence in multi-ancestry comparison and replication analysis."

4. Intuitively, the protein expression level could be different in case group and control group, using the summary statistic largely from control group may bias the causal estimates. The authors should provide rationale and discussions over this choice.

Reply: This is a very important point raised by reviewer 2. We totally agree that protein expression level could be different in case and control groups. As a case study, we have attempted to address this question by estimating the effect of cis-CRP variants in obese, diabetic patients and observed that the genetic effects of cis-CRP variants do vary across sub-groups of patients (<https://www.biorxiv.org/content/10.1101/2021.09.23.461369v1>). However, in this study we were focused on identifying causal proteins for disease prevention rather than disease progression or disease treatment. There are a few relevant points we would like to clarify here:

1. The outcome GWAS studies were in a case-control setting at one time point (rather than a case-only setting or a disease progression setting after disease onset). Using these data in a MR setting will estimate the causal effect of a protein on disease onset. In another word, our causal question is whether genetically predicted levels of proteins throughout life will increase or decrease risk of disease onset. Since we are not looking at progression, the protein level difference between cases and controls after onset of disease is not a major issue for our study.

2. Both exposure and outcome data were derived from the general population. For the exposure data, the pQTLs were derived from ARIC, which is a cohort of participants selected from the general population. For the outcome data, the outcome GWAS were derived from the meta-analysis of up to 19 biobanks, of which the vast majority were from the general population. Under the assumption that the prevalence of the tested disease was similar in the exposure and outcome samples (which is likely to be the case since both data were from the general population of the same ancestry group), our estimates should not be biased due to unbalanced number of cases and controls in the outcome data.

3. In addition, we did not select samples based on disease status. Therefore, selection bias / collider bias is unlikely to have a big impact on our results.

We agree with the reviewer that for future MR studies based on disease progression/treatment, it is very important to consider the differences in protein levels genetic effect modification between disease case and control groups.

Given the importance of this comment, we have added the following paragraph to the discussion section:

"In this study, the outcome data were derived from case-control GWAS studies. Therefore, the MR conducted here only estimated the causal effect of proteins on disease prevention rather than disease progression⁵⁶. In addition, the exposure and outcome data in this study were derived from the general population and the same ancestry group. Our MR estimates were unlikely to be biased due to the unbalanced number of cases and controls. However, future disease progression studies using case-only data will create new challenges. Differences in protein levels and genetic interactions across disease status will be important issues to be considered for instrument selection⁵⁷. Collider bias will be another key consideration, which has been discussed in recent papers^{58,59}. Some novel genetic epidemiology methods such as the Slope-Hunter approach⁶⁰ have been developed to detect and adjust potential collider bias introduced by disease progression data".

5. In the data availability section, I cannot find download links for summary data used.

Reply: Thank you for raising this question. The summary statistics from GBMI have recently been released to the public. We have added the download link to the data availability section.

Reviewer #3: I want to congratulate the authors for this very thorough analysis that aim to identify "putative" causal genes on a series of medical conditions with high unmet clinical need. The group of authors have extensive track record in this area. This group once more takes the field of MR one step forward. Below, some comments with the goal of improving the clarity and quality of this exceptional work.

Reply: Thanks for the kindly words from reviewer 3. The comments are very helpful further strengthen the manuscript.

Introduction:

1. First paragraph of this paper is described towards "pharmacogenetics", which though also involves drugs and genes is slightly different from the authors are attempting to do. To use human genetics to identify and validate drug targets for a wide range of disorders. Indeed the Ref #1 is about Pharmacogenetics (PGX), please modify the paragraph 1 and ref-1 accordingly. There are obvious reasons for using multi-ancestry for drug-discovery. e.g. LOF mutation in PCSK9 from African-American (NEJM) lead the way for drug-target discovery.

Reply: Thank you for the suggestion and sharing this nice NEJM paper as an example. We have reshaped the first paragraph of the manuscript and modify Reference 1 to the PCSK9 loss of function mutation from African Americans.

"The efficacy of drugs is typically evaluated in one or a small number of populations during phase 3 clinical trials. Utilising multi-ancestry data for drug discovery has several obvious benefits. First, ancestry-specific data may lead to novel drug target discovery. For example, a loss-of-function mutation in PCSK9 was first identified using sequencing data from African ancestries¹. Second, genetic tools provide a cost-effective way to understand whether drug targets may have similar or different effects across ancestries, which could help to improve the generalisability of drug interventions across ancestries."

2. Conceptual clarification: given the resources they are using (outcomes being prevalent plus incident (could not see this but is usual GWAS?)): This type of analysis is only relevant to identify drug-targets for primary prevention of the conditions described in this manuscript. Drug target identification for 2ry or acute management requires a different setting, which I assume some of the co-authors are well aware.

Reply: Thank you for helping us to clarify this concept. The reviewer is totally right that our study used case-control GWAS as outcomes, therefore addresses disease prevention rather than disease progression (this has been mentioned in reply to Q4, reviewer 2 as well). We have worked through the manuscript and made it clear that we prioritised drug targets for disease prevention in this study.

In the abstract:

"Our results highlight the value of proteome-wide MR in informing the generalisability of drug targets for disease prevention across ancestries and illustrate the value of multi-cohort and biobank meta-analysis of genetic data for drug development."

In Introduction:

"identified potential multi-ancestry and ancestry-specific causal proteins that prevent disease onset."

In Results:

"We undertook two-stage (discovery and validation) MR and sensitivity analyses to systematically evaluate evidence for the putative causal effects of 1,311 plasma proteins on the eight diseases in African ancestry and separately for 1,310 proteins on the same eight diseases in European ancestry. Of note, since the outcome data we used here are from case-control GWAS, the protein-disease pairs identified in this study are potential targets for disease prevention rather than treatment."

In Discussion:

"By estimating 21,470 putative causal effects of proteins on preventing relevant diseases in European and African ancestries, we found evidence for potential causal effects in 45 and seven protein-disease pairs in the two ancestries respectively."

3. With a few exceptions authors talk of "causal" evidence through the entire manuscript. I suggest they temper their language and use "putative" causal. This will match more closely with the large battery of analyses they conducted, which is clear evidence of the various assumptions and limitations of the strong strategy they applied. Indeed some of the assumptions of IV are unverifiable, something the authors are well versed.

Reply: Thank you for this nice suggestion. This is indeed a very important point to describe our findings. In this revision, we refined the term "causal" to "putative causal" through the whole manuscript. Some examples have been shown in Reply to Q2, reviewer 3.

Results /methods:

4. Clarification of the decision making process: Through the manuscript the authors used word to denote some categories/degree of credibility in their findings. Some examples I identified are: (a) Fig 6. they talk of "robust/ moderate". (b) Little evidence of horizontal pleiotropy. Please clarify these statements to secure reproducibility of their findings/strategy.

Reply: Thank you for the comments about decision making process. Given we have conducted a set of analyses, some of them have specific selection criteria, e.g. Figure 6, genetic evidence robust / moderate. We agree it is particularly important to clarify the decision-making process for each analysis. Therefore, we carefully checked all analyses and made the following table (a new table Table S19) to present the selection criteria for each of them:

Table S19. Selection criteria for each analysis included in this study

Analysis type	Selection criteria
Discovery	MR FDR<0.05, Coloc PP>0.7, Egger intercept P>0.05, Steiger filtering P>0.05
Replication	MR FDR<0.05, Coloc PP>0.7, Egger intercept P>0.05, Steiger filtering P>0.05
Sex-specific analysis	MR FDR<0.05, pair-wise Z-score P males/females <0.05, Egger intercept P>0.05, Steiger filtering P>0.05
Trans-ancestry comparison	MR FDR<0.05 in trans-ancestry comparison, MR P<0.05 in replication, Coloc PP>0.7, Egger intercept P>0.05, Steiger filtering P>0.05
Triangulation	
Robust genetic evidence	MR FDR<0.05 in trans-ancestry comparison, MR FDR<0.05 in replication, Coloc PP>0.7, Egger intercept P>0.05, Steiger filtering P>0.05
Moderate genetic evidence	MR FDR<0.05 in trans-ancestry comparison, MR P<0.05 in replication, Coloc PP>0.7, Egger intercept P>0.05, Steiger filtering P>0.05
Robust observational evidence	Observational association FDR<0.05 and showed same direction of effect as MR estimate Observational association FDR<0.05 or showed same direction of effect as MR estimate

Moderate observational evidence

Robust overall evidence Robust evidence in one type and at least moderate evidence in the other type

Moderate overall evidence Moderate genetic evidence or observational evidence did not reach the criteria of a moderate evidence level.

Re the question about evidence level in triangulation analysis, the protein-disease pairs listed in

Table S18 had passed a stringent selection criterion of MR $FDR < 0.05$ in trans-ancestry comparison, colocalization probability $> 70\%$, MR $P < 0.05$ in replication and passed sensitivity analyses (including MR Egger intercept test and Steiger filtering). To be conservative, we noted these protein-disease pairs as having “moderate” genetic evidence. On top of these criteria, if a proteindisease

pair also passed MR $FDR < 0.05$ in the replication, then it is well replicated and noted as “robust” genetic evidence. The robust observational evidence was defined as those pairs with $FDR < 0.05$ and showing the same direction of effect as MR estimate. Moderate observational evidence was defined as those pairs with either $FDR < 0.05$, or showing the same direction of effect as the MR estimate. Integrating the genetic and observational evidence, if a pair showed robust evidence in one type of evidence and at least moderate in the other type of evidence, we noted it as robust overall evidence. Otherwise, the overall evidence will be moderate. The eight proteindisease pairs with robust overall evidence were presented in Figure 6.

Re the question about little evidence of horizontal pleiotropy, this means the MR estimate showed a P value of the MR Egger intercept term > 0.05 in our study.

In addition, to avoiding misleading interpretation of our drug target prioritization, we have added the following disclaimer in the Results section:

“It is important to note that our drug target prioritisation was based on the use of a P value threshold

for MR evidence ($FDR < 0.05$ based on gIVW estimates), which we simply use as a heuristic for highlighting putative causal effects worthy of follow-up. Investigations of results can therefore apply more (or less) stringent thresholds by filtering the protein-disease effects downloadable from our web browser (<https://epigraphdb.org/multi-ancestry-pwmmr/>).”

5. pQTL sources/replication: I wonder if possible to describe the overlap among the different sources of pQTLs. Plus, as I Somascan user, I am aware those sources are from different versions, which add some degree of technical difficulty for interoperability. Can the authors comment on this? specially as the pQTLs used for discovery and replication comes from wide variety of SOMAscan version. A question of clarification: is the use of alternative sources of pQTL a replication or a sensitivity analysis? To this reviewer the sensitivity analysis seems more appropriate (at least I am missing something).

Reply: Thank you for raising this question. We agree it is a useful sensitivity analysis to compare pQTL effects across different version of the SOMAscan platform. In this revision, we first searched literature that compared pQTLs from different version of SOMAscan platforms. The DeCODE pQTL paper showed that 83% of the tested pQTLs overlapped between DeCODE pQTLs (which used SOMAscan v4) and Sun et al pQTLs (which used SOMAscan v3). In another study, 64% (867 of 1356 pQTLs) overlapped between Fenland pQTLs (SOMAscan v4) and Sun et al pQTLs. This means pQTL overlap was moderate across different SOMAscan platforms.

To check pQTL overlap in our study, we compared the instruments we used in the replication analysis for European ancestry (mainly from Sun et al and Emilsson et al. using SOMAscan v3) with the recent released summary statistics from DeCODE proteome GWAS (SOMAscan v4; same version as ARIC data but completely independent samples). The results suggested that 98.5% (270 of 274 pQTLs) showed the same direction of effect. 40.0% (110 of 275 pQTLs) had overlapping 95% confidence intervals for the pQTL effects (Table S4). This suggested that the direction of the pQTL effects across SOMAscan versions are almost the same, but the actual pQTL effect sizes could be different.

To present the above findings in the revision, we have changed the word “replication” to “validation” through the manuscript and Figure 1. We have also included the pQTL comparison across different version of SOMAscan platform in Table S4 and added the following sentences in the discussion:

“Fifth, for the validation analysis in European ancestry, the pQTLs were obtained from studies using SOMAscan v3 platform, whilst the discovery pQTLs were obtained from studies using SOMAscan v4 platforms. We observed some levels of departure for pQTL effect sizes across these studies (see Table S4), but whether this is due to version difference of SOMAscan or other differences needs further investigation.”

6. Quality of the Phenotypes: there is no mention as a weakness that the phenotypes used are just Phecodes. These are just ICD-only phenotypes without "clinical chart review" adequate for this type of large-scale discovery process, but clearly not ideal if the implications are to initiate/continue discovery programs and trials with large-financial implications, please acknowledge this as a limitation.

Reply: Thank you for this recommendation. We have added this limitation as follows:

“Fourth, the disease cases were defined using Phecodes such as ICD10 codes. There is a potential limitation that the quality of definition of the disease cases may not be as reliable as clinically diagnosed

cases. According to the comparison conducted in the GBMI flagship and disease specific GWAS papers, the genetic effects for clinically-diagnosed IPF₅₆, VTE₄₅ and AAA₂₂ were similar compared to those for the Phecode-defined data. However, the genetic effect of Asthma showed quite different effect estimates using the two definitions⁵⁷.”

7. Findings with $p < 0.05$ but $FDR > 5\%$: please remove these findings. This goes against the very strict steps/approach authors used to minimize any false positive findings.

Reply: Thank you, we have removed these results in the discovery and replication analyses section of the main text.

8. Triangulation of evidence: I like the terminology and endorse the use of complementary sources of evidence. However, I have some strong reservations to the ones used by the authors. Observational: some of the authors of this paper have wrote extensively about the limitations of observational evidence as the reason for MR to exist. Now I am surprise how now they reintroduce this evidence, and even categorize as "moderate" only after logistic regression with age and sex as co-variates. This reviewer disagree with this statement and propose to exclude this. Also please clarify what exactly is the "extra" evidence provided by trial.

Reply: We thank reviewer 3 for raising this important question. We strongly agree with the reviewer that calling the observational results “moderate evidence” is not a good approach and may be misleading. Therefore, we have removed such statements from the main text, Figure 6 and Table S18. Instead, we only included the original observational results in Table S16 and S18. In addition, for the genetic evidence, we consider the words “robust” and “moderate” can be misleading too. We have also removed these from Figure 6.

In addition, we agree with the reviewer that MR aims to identify putative causal effects, whilst observational analysis aims to identify associations, which are two distinguished concepts in epidemiology. We consider the concept of triangulation as a practice to obtain more reliable answers to research questions through integrating results from several different approaches, where each approach has different sources of potential bias that are unrelated to each other (Lawlor et al., 2016). In this study, we attempted to integrate results from different approaches, including MR/colocalization and observational associations. They also have very different potential biases, e.g. in MR, horizontal pleiotropy is a major source of bias, whilst in observational analysis confounding and reverse causality are major sources of biases. Given these differences of the two approaches, our aim here for triangulation is that, if the results of different study designs all point to the same conclusion, this strengthens the confidence level of the findings. We are therefore not attempting to suggest that the MR and observational estimates are equivalent, but that if there is a true effect of a protein, we might expect to see evidence for both causal estimates and observational

associations. Clinical trials provide a third study design, subject to different biases from MR and observational studies (e.g. performance and detection bias). The “extra” evidence provided by this study type is simply an extension of the triangulation mentioned above.

Figure 6.

9. Novelty: can the authors clarify the process to declare novelty? Do they use other MR - GWAS-pQTLs recently published or GWAS-catalogue.

Reply: Thank you for raising this question. In this revision, we conducted a literature review and database search for the eight prioritized targets illustrated in figure 6B with existing pQTL GWAS and MR findings and categorized the results into three types:

1. Existing drug/target: reported in pQTL studies (as protein-disease pair), where Open Targets drug score > 0 (which refers to protein-disease pairs with clear pQTL evidence and some drug trial evidence).
2. Novel drug target: reported in pQTL studies (protein-disease pair), where Open Targets

drug score=0 (which refers to protein-disease pairs with some genetic evidence, but with little drug trial evidence).

3. Novel protein-disease pair: not reported in previous pQTL studies, where Open Targets drug score=0 (which refers to protein-disease pairs with little genetic evidence, and little drug trial evidence)

For the reported pQTL studies, we searched for ten existing pQTL GWAS and MR studies, which include: Fenland (Pietzner et al., 2021), DeCODE (Feringstad et al., 2021), INTERVAL (Sun et al., 2018b), Emilsson et al. (Emilsson et al., 2018)(Emilsson et al., 2020), Folkersen et al. (Folkersen et al., 2020), Yao et al. (Yao et al., 2018), Zhang et al. (Zhang et al., 2021), Zheng et al. (Zheng et al., 2020) and Suhre et al (Suhre et al., 2017).

We categorized our top findings based on the above criteria (see Table below):

1. Novel drug targets: we found 5 of the 8 prioritized targets are novel drug targets with existing genetic evidence but with little trial evidence: SERPINE2 on VTE; F11 on VTE; IL7R on Asthma; ABO on VTE and TNFSF12 on Asthma.

2. Novel protein-disease pairs: two pairs, SERPINF1 on Asthma and KLKB1 on VTE could be considered as a novel protein-disease pair since little pQTL or trial evidence have been identified.

3. Existing drug/targets: ACE inhibition has been proposed as drug target to treat COPD in a Phase IV trial (NCT01014338). Our study found that ACE inhibition is more likely to be effective in African ancestry.

To better present these results, we have added a new section in the Results and have included a new column in Figure 6B to show the novelty of our findings.

“To define novelty, we further compared the eight prioritised protein-disease pairs with existing pQTL GWAS and MR findings reported in ten proteome studies^{4,5,6,7,8,18,20,63,64,65}. Two of the pairs, SERPINF1 on Asthma and KLKB1 on VTE, were not reported in any of the pQTL studies and are not under clinical investigation yet, which we have therefore suggested are completely novel protein-disease pairs identified in this study. Five pairs were reported in previous pQTL studies but are not under clinical investigation yet, which we have suggested as novel drug targets. The pair ACE on COPE was not reported in any pQTL studies but is under a Phase IV trial, therefore we considered this to be an existing drug/target with potential to be validated in African ancestry (Figure 6B; Table S18). Our study provides evidence to support formal investigations of these protein-disease pairs in future clinical trials.”

Table. Define novelty of the top MR findings.

Protein	Outcome	Ancestry specificity	Open Target drug score	Reported in existing pQTL papers	Type of Novelty
SERPINE2	VTE	Trans-ancestry	0	Decode	Novel drug target
KLKB1	VTE	European-specific	0	Not reported	Novel drug target
F11	VTE	European-specific	0	Fenland; Decode; Zheng	Novel drug target
IL7R	Asthma	European-specific	0	Sun	Novel drug target
ABO	VTE	Trans-ancestry	0	Fenland; Decode	Novel drug target
SERPINF1	Stroke	African-specific	0	Not reported	Novel protein-disease pair
TNFSF12	Asthma	European-specific	0	Decode	Novel drug target
ACE	COPD	African-specific	0.61	Not reported	Existing drug/target Validation in African

10. LD-check: I really like this, statistical colocalization does not remove confounding by LD. However, can the authors clarify what is exactly the trait/s they checked against? I was under

the assumption they were checking for the same or related trait as the one the MR discovered, please clarify. Also please clarify how did you solve the problem of GWAS using different terminology for phenotypes.

Reply: Thank you for the positive feedback about LD check. We agree with the reviewer that colocalization is a way to detect confounding by LD but it is hard to totally exclude such situation. Part of the reason is methodological, e.g. the influence of multiple signals in the same region. For the LD check analysis we developed, it is an approach to check the LD structure of the outcome data in a test region. For a given pQTL, we assumed it is the sentinel variant and selected a genomic region around this variant (e.g. a 1Mb window). We looked up the genetic signals of the tested outcome within this region, where the outcome is the same phenotype/disease what we tested in the MR analysis. For all genetic signals with $P < 1e-5$ in the outcome data, we checked their pairwise

LD with the centinal pQTL variant. If an outcome signal was in strong LD with the sentinel pQTL ($r_2 > 0.7$ in relevant population), then we consider the outcome variant was a shared signal with the pQTL in this region. The advantage of this approach is that it allows multiple pQTL signals and/or multiple outcome signals in the region. If any pair of the pQTL-outcome signal was “LD checked”, then they are likely a shared signal in the region. Although we called this an approximate colocalization approach, we consider LD check as a more flexible and computationally friendly approach.

With regard to terminology for phenotypes, the outcomes used for the LD-check analysis are the same as the outcomes for MR and colocalization analyses. Therefore, the terminology of phenotypes is not an issue in this case.

We really appreciated that the reviewer’s suggestion about checking LD between related traits as the one the MR discovered, which can be a useful extension of this method. We will consider this extension more carefully and aim to develop it in a methodological paper.

11. Recommendations: I will assume that other groups will have slightly different variations of the recommendations. I have a few to start with: (a) The authors of MR should clearly specify what is the target trial they are attempting to emulate. (b) efficacy outcomes should be those used in the trial they are attempting to emulate or as close as possible. (c) MR can also inform on on-target safety events (reason as to which 30% of the new drug-discovery programs failed). I will call (a) and (b) in-silico trials. (d) horizontal pleiotropy: for cis-MR where is very rare to have >3 instruments or ideally 10 to run MR Egger, HP is the main problem. Despite that this is a topic whose conditions are clearly defined, the approaches most people take (including this group) are approximations (e.g. instrument >5 proteins, or test for heterogeneity) are not indeed test for horizontal pleiotropy.

Reply: Thank you for all the nice suggestions about the recommendation section. We do agree that proteome MR is a big research area. Although we put some efforts in this area, our recommendations are somewhat subjective and based on our own experience. For the four suggestions from the reviewer, we carefully considered and discussed amongst authors. Our replies are:

1. Point (a) and (b), thank you for comparing MR with in-silico trial, which is a very novel way of thinking. Although we called MR as a natural randomized trial, we still consider MR as a quite different concept to an in-silico trial. An in-silico trial is a computationally simulated clinical intervention, where the ‘virtual’ patients will be given a ‘virtual’ treatment to check the efficacy. So, it is a computational simulation/prediction model with clear aims. In contrast, when conducting MR, we are using real data from the population to do causal inference, which is a kind of observational analysis. But we agree with the reviewer that there may be a way to integrate these two approaches together, e.g. in

designing future in-silico trials, we could consider some MR evidence, e.g. target effect on a disease, as parameters to improve prediction/simulation accuracy.

2. Point (B), we agree with the reviewer that, in a MR setting of one target on one indication, the exposure and outcome need to be set as close as possible, e.g. in the same candidate population with similar disease status, age, sex etc. This is what we are aiming to achieve in future MR design/methods development.

3. Point (C), again a very important point, which need to be included in the recommendation.

4. Point (D), yes, we totally agree with the reviewer that MR Egger is not the ideal case for cis-MR. In fact, horizontal pleiotropy is a major issue for all types of MR, the situation is harder to solve for cis-MR. Besides the widely used methods we applied, some other methods such as MR-PheWAS, MR RAPS, MR-TRXY and the contamination mixture method may bring some new opportunities to adjust for pleiotropy in cis-MR.

We have added some more recommendations based on point (B), (C) and (D).

"In comparisons of MR and RCT, the exposure and outcome of the two approaches need to be as close as possible, for example, the outcome definition need to be similar (e.g. similar population disease status, age and sex)."

"In addition to novel target identification and drug repurposing, MR can also inform on-target safety events of drug targets, potentially addressing some of the approximately 30% of new drugdiscovery programs that fail due to safety considerations."

"For sensitivity analyses, given the limited number of cis-acting pQTLs for each protein, it is difficult to apply classic sensitivity methods, such as weighted median and mode estimator methods to test for MR assumptions. The following methods are recommended for current and future proteomics MR:

...

Some recent methods such as CAUSE account for correlated and uncorrelated pleiotropy using genome-wide summary statistics⁸⁰, but are not optimized for analysis in a specific genomic regions. With more efforts in methods development (e.g. a CAUSE based method in the cis-acting region) and evaluation of other methods such as contamination mixture methods⁸¹ and MR locus⁸², such approaches may offer the potential to estimate level of pleiotropy in the near future.

References

- Emilsson, V., Ilkov, M., Lamb, J.R., Finkel, N., Gudmundsson, E.F., Pitts, R., Hoover, H., Gudmundsdottir, V., Horman, S.R., Aspelund, T., et al. (2018). Co-regulatory networks of human serum proteins link genetics to disease. *Science* <https://doi.org/10.1126/science.aaq1327>.
- Emilsson, V., Gudmundsdottir, V., Ilkov, M., Staley, J.R., Gudjonsson, A., Gudmundsson, E.F., Launer, L.J., Lindeman, J.H., Morton, N.M., Aspelund, T., et al. (2020). Human serum proteome profoundly overlaps with genetic signatures of disease.
- Ferkingstad, E., Sulem, P., Atlason, B.A., Sveinbjornsson, G., Magnusson, M.I., Styrnisdottir, E.L., Gunnarsdottir, K., Helgason, A., Oddsson, A., Halldorsson, B.V., et al. (2021). Large-scale integration of the plasma proteome with genetics and disease. *Nat. Genet.* 1–10. <https://doi.org/10.1038/s41588-021-00978-w>.
- Folkersen, L., Gustafsson, S., Wang, Q., Hansen, D.H., Hedman, Å.K., Schork, A., Page, K., Zhernakova, D.V., Wu, Y., Peters, J., et al. (2020). Genomic and drug target evaluation of 90 cardiovascular proteins in 30,931 individuals. *Nature Metabolism* 2, 1135–1148. <https://doi.org/10.1038/s42255-020-00287-2>.
- Gkatzionis, A., Burgess, S., and Newcombe, P.J. (2021). *Statistical Methods for cis-Mendelian Randomization*.

- Lawlor, D.A., Tilling, K., and Davey Smith, G. (2016). Triangulation in aetiological epidemiology. *Int. J. Epidemiol.* 45, 1866–1886. <https://doi.org/10.1093/ije/dyw314>.
- Morrison, J., Knoblauch, N., Marcus, J.H., Stephens, M., and He, X. (2020). Mendelian randomization accounting for correlated and uncorrelated pleiotropic effects using genome-wide summary statistics. *Nat. Genet.* 52, 740–747. <https://doi.org/10.1038/s41588-020-0631-4>.
- Pietzner, M., Wheeler, E., Carrasco-Zanini, J., Cortes, A., Koprulu, M., Wörheide, M.A., Oerton, E., Cook, J., Stewart, I.D., Kerrison, N.D., et al. (2021). Mapping the proteogenomic convergence of human diseases. *Science* eabj1541. <https://doi.org/10.1126/science.abj1541>.
- Sanderson, E., Glymour, M.M., Holmes, M.V., Kang, H., Morrison, J., Munafò, M.R., Palmer, T., Schooling, C.M., Wallace, C., Zhao, Q., et al. (2022). Mendelian randomization. *Nature Reviews Methods Primers* 2, 1–21. <https://doi.org/10.1038/s43586-021-00092-5>.
- Suhre, K., Arnold, M., Bhagwat, A.M., Cotton, R.J., Engelke, R., Raffler, J., Sarwath, H., Thareja, G., Wahl, A., DeLisle, R.K., et al. (2017). Connecting genetic risk to disease end points through the human blood plasma proteome. *Nat. Commun.* 8, 14357. <https://doi.org/10.1038/ncomms14357>.
- Sun, B.B., Maranville, J.C., Peters, J.E., Stacey, D., Staley, J.R., Blackshaw, J., Burgess, S., Jiang, T., Paige, E., Surendran, P., et al. (2018a). Genomic atlas of the human plasma proteome. *Nature* 558, 73–79. <https://doi.org/10.1038/s41586-018-0175-2>.
- Sun, B.B., Maranville, J.C., Peters, J.E., Stacey, D., Staley, J.R., Blackshaw, J., Burgess, S., Jiang, T., Paige, E., Surendran, P., et al. (2018b). Genomic atlas of the human plasma proteome. *Nature* 558, 73–79. <https://doi.org/10.1038/s41586-018-0175-2>.
- Yao, C., Chen, G., Song, C., Keefe, J., Mendelson, M., Huan, T., Sun, B.B., Laser, A., Maranville, J.C., Wu, H., et al. (2018). Genome-wide mapping of plasma protein QTLs identifies putatively causal genes and pathways for cardiovascular disease. *Nat. Commun.* 9, 3268. <https://doi.org/10.1038/s41467-018-05512-x>.
- Zhang, J., Dutta, D., Köttgen, A., Tin, A., Schlosser, P., Grams, M.E., Harvey, B., Yu, B., Boerwinkle, E., Coresh, J., et al. (2021). Large Bi-Ethnic Study of Plasma Proteome Leads to Comprehensive Mapping of cis-pQTL and Models for Proteome-wide Association Studies.
- Zheng, J., Haberland, V., Baird, D., Walker, V., Haycock, P.C., Hurle, M.R., Gutteridge, A., Erola, P., Liu, Y., Luo, S., et al. (2020). Phenome-wide Mendelian randomization mapping the influence of the plasma proteome on complex diseases. *Nat. Genet.* 52, 1122–1131. <https://doi.org/10.1038/s41588-020-0682-6>.

Referees' report, second round of review

Reviewer#1

I thank the authors for addressing all my comments. I have no further remarks.

Reviewer#2

Comments enter in this field will be shared with the author; your identity will remain anonymous.

Thanks for addressing my comments. Here I still have two additional comments.

1. The selection of IVs can substantially affect the performance of MR methods, for instance, Winner's curse in addition to weak IVs. To minimize potential bias due to the winner's curse, Zhao et al. (2019) suggested selecting the IVs first using a third independent sample. On the

other hand, the use of weak IVs can substantially improve statistical power (Cheng et al., 2020; Yuan et al., 2022). Authors need to discuss these issues.

2. It is widely observed that CAUSE suffers from conservatism and loss of power. cML-MA, GRAPPLE and MR-APSS all can estimate causal effects in the presence of both uncorrelated and correlated horizontal pleiotropy. It will be more persuasive to compare those methods with reasonable power. The field of MR methods develops very fast in last few years. gIVW and gEgger have been developed without consideration of violations of three key MR assumptions. Methods that consider correlated horizontal pleiotropy can further account for reverse causation at least partially. The wide use of IVW/Egger-based methods may be attributed to their simple models and ease of implementations.

Cheng, Q., Yang, Y., Shi, X., Yeung, K. F., Yang, C., Peng, H., & Liu, J. (2020). MR-LDP: a two-sample Mendelian randomization for GWAS summary statistics accounting for linkage disequilibrium and horizontal pleiotropy. *NAR genomics and bioinformatics*, 2(2), lqaa028.

Yuan, Z., Liu, L., Guo, P., Yan, R., Xue, F., & Zhou, X. (2022). Likelihood-based Mendelian randomization analysis with automated instrument selection and horizontal pleiotropic modeling. *Science advances*, 8(9), eabl5744.

Zhao, Q., Chen, Y., Wang, J., & Small, D. S. (2019). Powerful three-sample genome-wide design and robust statistical inference in summary-data Mendelian randomization. *International journal of epidemiology*, 48(5), 1478-1492.

Reviewer#3

Many thanks for your v detailed responses. I have no further comments.

Authors' response to the second round of review

Reviewer #1: I thank the authors for addressing all my comments. I have no further remarks.

Reply: Thanks for the comments from Reviewer 1; we appreciate their detailed comments and helpful suggestions. Hope this manuscript can be a start point of developing multiancestry drug target Mendelian randomization.

Reviewer #2: Thanks for addressing my comments. Here I still have two additional comments.

Reply: Thank you for the nice suggestions from an MR method point of view. Both comments are very important, and have helped us to improve the robustness of our manuscript and findings.

1. The selection of IVs can substantially affect the performance of MR methods, for instance, Winner's curse in addition to weak IVs. To minimize potential bias due to the winner's curse, Zhao et al. (2019) suggested selecting the IVs first using a third independent sample. On the other hand, the use of weak IVs can substantially improve statistical power (Cheng et al., 2020; Yuan et al., 2022). Authors need to discuss these issues.

Reply: Thank-you for raising this important point. We agree with Reviewer 2 that Winner's curse will become a serious issue when there is sample overlap between the two samples used. We also agree that a three-sample MR analysis will be very valuable once more pQTL resources have been created and released. We have added the references Reviewer 2 recommended to the main text. Furthermore, a recent MedRxiv paper systematically

estimated the influence of sample overlap, Winner's curse and weak instrument bias through different choices of study samples (Sadreev et al., 2021). This study suggested that winner's curse incurs substantial overestimation of effect sizes in a mean of 35% of discovered associations per trait in UK Biobank. As the simulation showed in this study, a genome-wide instrument selection approach will provide more reliable MR estimates compared with a "significant SNP" approach with existence of sample overlap and Winner's curse (Figure 1 in Sadreev et al., 2021). In addition, this paper suggested that the direction of the bias is towards the null when exposure and outcome samples are independent. In our study, the samples of our exposures (ARIC) and outcomes (GBMI) are completely independent. Therefore, the influence of Winner's curse is towards the null. As Reviewer 2 suggested, we have added a new paragraph in the discussion to explain the importance of considering Winner's curse/sample overlap for MR analysis.

"winner's curse of GWAS estimates will bias the downstream MR estimates towards the null when exposure and outcome samples are independent. A recent study suggested that
Response to Reviewers

winner's curse incurs substantial overestimation of effect sizes in a mean of 35% of discovered associations per trait in UK Biobank⁷². The same study suggested that a threesample MR setting⁷³ using two independent exposure datasets may reduce the influence of winner's curse. In addition, some recent studies have suggested that the use of genomewide instruments and inclusion of weak instruments may improve the reliability of MR estimates and increase power compared to a classic "significant SNP" approach^{72,74,75}."

2. It is widely observed that CAUSE suffers from conservatism and loss of power. cML-MA, GRAPPLE and MR-APSS all can estimate causal effects in the presence of both uncorrelated and correlated horizontal pleiotropy. It will be more persuasive to compare those methods with reasonable power. The field of MR methods develops very fast in last few years. gIVW and gEgger have been developed without consideration of violations of three key MR assumptions. Methods that consider correlated horizontal pleiotropy can further account for reverse causation at least partially. The wide use of IVW/Egger-based methods may be attributed to their simple models and ease of implementations.

Reply: Thank you for the suggestions from Reviewer 2 re considering other MR methods that take into account both uncorrelated and correlated horizontal pleiotropy. In this revision, we have evaluated the three suggested methods, cML-MA, GRAPPLE and MRAPSS, on the eight top MR findings listed in Fig6B. In applying the same analysis to both European and African ancestries, we ended up conducting 15 analyses for each method.

cML-MA (constrained maximum likelihood and model averaging) is a MR method that is robust to invalid instrument variables (IVs) with uncorrelated or correlated pleiotropic effects. cML-MA has two settings, one using data perturbation (cML-MADP), and one not (cML-MA). In order to get more reliable results, we performed cMLMA-DP with data perturbation and set 100 random start points. The results suggested that all protein-disease pairs with robust gIVW evidence also showed robust MR evidence using cML-MA-DP, with 12 of the 15 pairs showing stronger power using cML-MA-DP. 13 of the 15 MR estimates also showed the same direction of effect between gIVW and cML-MA-DP, while the other 2 with different direction of effect both showed litter MR evidence. The only drawback is that the cML-MA-DP method is computationally expensive. In an example using a set of 56 instruments for the

SERPINE2 protein, the cML-MA-DP method took 7 mins to finish (100 random start point), while the cML-MA method used 30 seconds to finish. Both models are slower than the gIVW method, which only need 2-3 seconds to finish.

GRAPPLE (Genome-wide mR Analysis under Pervasive PLEiotropy) is a comprehensive framework for analyzing the causal effect of target risk factors with heterogeneous genetic instruments and identifying possible pleiotropic patterns from data. This method requires two files for the risk factor, one for selection files and the other for exposure files. Selection files are used to guard against instrument selection bias and exposure files are used for MR analysis. In this study, we used GWAS data of one protein from one ancestry as selection files and GWAS of the same protein in the other ancestry as exposure files. Since we don't have data for the *IL7R* protein for African ancestry, we conducted analyses on 14 pairs using GRAPPLE. Comparing GRAPPLE results with gIVW results, 12 of the 14 protein-disease pairs showed consistent directionality. For the 2 inconsistent pairs, *KLKB1* on VTE (African ancestry) and *SERPINF1* on Stroke (European ancestry) showed different directionality but little MR evidence, where *KLKB1* on VTE (African ancestry) was the same as ML-MA-DP suggested. 8 of the 14 protein-disease pairs showed consistent MR evidence. For the 6 pairs with inconsistent MR evidence, 2 of them only showed robust MR evidence using gIVW, while 4 of them only showed robust MR evidence using GRAPPLE. We consider such inconsistent MR evidence between the two approaches could be caused by the selection files we used for GRAPPLE. As mentioned above, the selection files and exposure files are from two different ancestries. Although the developer of GRAPPLE suggested that selection files and exposure files can come from different ancestries, we still considered that genetic association signals of the same SNP could be influenced by different allele frequencies, LD structures and statistical power across ancestries. In addition, the pleiotropic effect variance estimates τ^2_{hat} is relatively small among all the test results in GRAPPLE, indicating that our top findings are rarely affected by uncorrelated or correlated pleiotropic effects. In summary, GRAPPLE used a three-sample MR setting, which increased its robustness to deal with weak instrument bias and influences of Winner's curse. However, it required two sets of exposure data, which is still difficult to obtain for pQTL analyses, especially for African data. Our comparison also suggested that more simulations/evaluations are needed before applying multi-ancestry exposures in the three-sample MR setting (such as GRAPPLE).

MR accounting for Pleiotropy and Sample Structure (MR-APSS) is another unified approach that uses genome-wide summary statistics to estimate causal effects. However, there are two reasons that stopped us from applying this method here. First, this method implements part of LD score regression, which maps genetic variants with HAPMAP3 reference panel. Since our original genetic association data showed results in the cis-acting regions, after mapping to HAPMAP3 reference panel and LD clumping, we were not able to identify enough SNPs to fit the MR model. Second, this method uses LD scores provided by the developers of LD score regression software. However, only LD scores for European and East Asian populations were provided, and the LD scores for Africans were missing. Given half of the analyses in our study were conducted using data from African ancestry, we were not able to run MR-APSS directly without creating the LD scores for African ancestry. In summary, we consider MR-APSS

is a good approach using genome-wide summary statistics in European and East Asian ancestries, but its application for drug target MR in Africans needs additional LD score data from an appropriate population sample.

In summary, we compared gIVW, cML-MA, GRAPPLE and CAUSE from three angles: causal direction, statistical power, and computational speed. These comparisons are illustrated in the table below.

Method	Causal direction	Statistical power	Computation speed
gIVW	Reference	11/15 with MR evidence (reference)	fast
cML-MA	13/15 consistent	11/15 with MR evidence (100% consistent with gIVW)	relatively slow
GRAPPLE	12/14 consistent	11/14 with robust evidence (73% consistent with gIVW)	moderate
CAUSE	11/15 consistent	4/15 with robust evidence (100% consistent with gIVW)	moderate

To show these new results, we have created a new supplementary table (Table S19) and added the following paragraph in the Discussion section:

“Some recent methods such as CAUSE account for correlated and uncorrelated pleiotropy using genome-wide summary statistics⁸⁶, but are not optimized for analysis in a specific genomic region. Other methods such as cML-MA⁸⁹, GRAPPLE⁹⁰ and MA-APSS⁹¹ were developed to deal with correlated and uncorrelated pleiotropy. Based on our pilot comparison, cML-MA provided the best statistical power and accuracy among these methods in a drug target MR setting (see Table S19). With more efforts in methods development and evaluation of other methods such as contamination mixture method⁸⁷ and MR locus⁸⁸, such approaches may offer the potential to effectively estimate the level and impact of pleiotropy on causal estimates in the near future.”

We also agree with Reviewer 2 that a good number of novel MR methods have been developed recently. gIVW and gEgger showed a relaxed 3rd assumption of MR, e.g. assume either no pleiotropy or balanced pleiotropy. But these models still carefully considered the first and second assumption of MR. In contrast, CAUSE and other methods that consider correlated pleiotropy relax the first assumption of MR (e.g. including weak instruments in the model). Different MR methods therefore have their own advantages and disadvantages, highlighting the value of comparing their impact on the results as we have now done. It is also important to provide guidance on which type of MR method is most appropriate for a particular type of data and study design, and we hope our additional analyses will be useful for others considering these methods for drug target MR.

Reviewer #3: Many thanks for your v detailed responses. I have no further comments.

Reply: Thank you for the high-quality comments from Reviewer 3. Appreciated this.

Reference:

Hu, X., Zhao, J., Lin, Z., Wang, Y., Peng, H., Zhao, H., Wan, X., and Yang, C. (2022).

Mendelian randomization for causal inference accounting for pleiotropy and sample structure using genome-wide summary statistics. Proc. Natl. Acad. Sci. U. S. A. 119, e2106858119.

<https://doi.org/10.1073/pnas.2106858119>.

Sadreev, I.I., Elsworth, B.L., Mitchell, R.E., Paternoster, L., Sanderson, E., Davies, N.M.,

Millard, L.A.C., Smith, G.D., Haycock, P.C., Bowden, J., et al. (2021). Navigating sample overlap, winner's curse and weak instrument bias in Mendelian randomization studies using the UK Biobank.

Wang, J., Zhao, Q., Bowden, J., Hemani, G., Davey Smith, G., Small, D.S., and Zhang, N.R. (2021). Causal inference for heritable phenotypic risk factors using heterogeneous genetic instruments. *PLoS Genet.* 17, e1009575. <https://doi.org/10.1371/journal.pgen.1009575>.

Xue, H., Shen, X., and Pan, W. (2021). Constrained maximum likelihood-based Mendelian randomization robust to both correlated and uncorrelated pleiotropic effects. *Am. J. Hum. Genet.* 108, 1251–1269. <https://doi.org/10.1016/j.ajhg.2021.05.014>.